



# Polynomial chaos to efficiently compute the annual energy production in wind farm layout optimization

Andrés Santiago Padrón[1], Jared Thomas[2], Andrew P. J. Stanley[2], Juan J. Alonso[1], and Andrew Ning[2]

[1]Department of Aeronautics & Astronautics, Stanford University, Stanford, CA, 94305, USA
[2]Department of Mechanical Engineering, Brigham Young University, Provo, UT, 84602, USA

*Correspondence to:* Andrew Ning (aning@byu.edu)

**Abstract.**

In this paper, we develop computationally-efficient techniques to calculate statistics used in wind farm optimization with the goal of enabling the use of higher-fidelity models and larger wind farm optimization problems. We apply these techniques to maximize the Annual Energy Production (AEP) of a wind farm by optimizing the position of the individual wind turbines. The AEP (a statistic itself) is the expected power produced by the wind farm over a period of one year subject to uncertainties in the wind conditions (wind direction and wind speed) that are described with empirically-determined probability distributions. To compute the AEP of the wind farm, we use a wake model to simulate the power at different input conditions composed of wind direction and wind speed pairs. We use polynomial chaos (PC), an uncertainty quantification method, to construct a polynomial approximation of the power over the entire stochastic space and to efficiently (using as few simulations as possible) compute the expected power (AEP). We explore both regression and quadrature approaches to compute the PC coefficients. PC based on regression is significantly more efficient than the rectangle rule (the method most commonly used to compute the expected power). With PC based on regression, we have reduced by as much as an order of magnitude the number of simulations required to accurately compute the AEP, thus enabling the use of more expensive, higher-fidelity models or larger wind farm optimizations. We perform a large suite of gradient-based optimizations with different initial turbine locations and with different numbers of samples to compute the AEP. The optimizations with PC based on regression result in optimized layouts that produce the same AEP as the optimized layouts found with the rectangle rule but using only one-third of the samples. Furthermore, for the same number of samples, the AEP of the optimal layouts found with PC is 1 % higher than the AEP of the layouts found with the rectangle rule.

## 1  Introduction

In 2015, wind energy growth accounted for almost half of the global electricity supply growth. In the United States, it accounted for 41 % of new power capacity, raising the wind energy supply to 4.7 % of the total electricity generated in 2015 and on target to reach 10 % by 2020 (U.S. Department of Energy, 2015; AWEA, 2016; GWEC, 2016). Most of the current and upcoming wind energy comes from large turbines (greater than 1 MW) situated in clusters—wind farms. A problem with putting turbines together in confined spaces is that they operate in the wakes of other turbines, i.e., in regions of reduced speed and increased



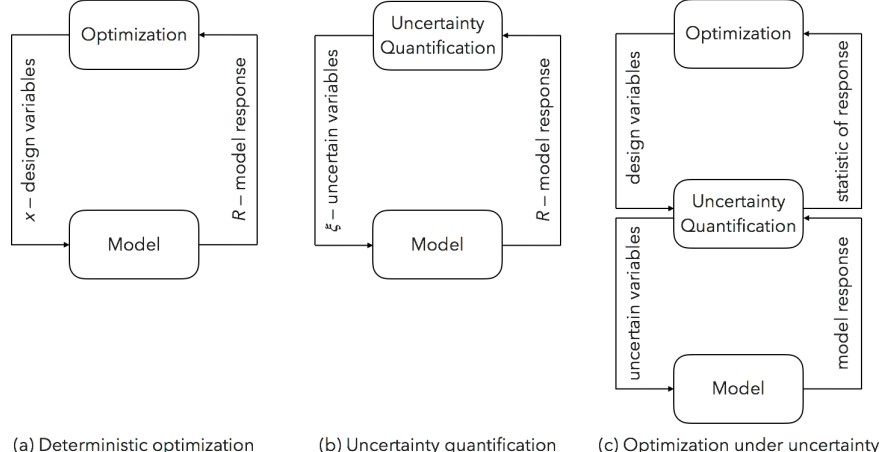

(a) Deterministic optimization  (b) Uncertainty quantification  (c) Optimization under uncertainty

**Figure 1.** Examples of applications that require many model evaluations. In deterministic optimization (a), we evaluate the model at different values of the design variables while searching for an optimum response. In uncertainty quantification (b), we query the model multiple times at different instances of the uncertain variables to generate an ensemble of responses from which we can compute statistics and probabilities of the model response. In optimization under uncertainty (OUU), we optimize a statistic. OUU is computationally expensive as it requires many model evaluations because of the nested loop in the problem formulation (c).

turbulence. This leads to an underproduction of power and decreased (10–20 %) energy output for the farm (Barthelmie et al., 2007, 2009; Briggs, 2013) when compared to ideal conditions. This loss in energy capture results in millions of dollars of loss for operators and investors and increased economic uncertainty for new installations. Many current wind farms have grid-like layouts, where wind turbines are aligned in rows, which further exacerbate the wake losses. By optimizing the layout of the

wind farm, the wake losses can be minimized, with a corresponding increase in energy production and revenue.

Wind farm optimization is a complex, multi-disciplinary and high-dimensional problem. The wind farm may contain dozens or even hundreds of wind turbines, where each turbine may be parameterically described using several design variables. Furthermore, the wind conditions (wind direction, wind speed, wind turbulence, etc.) are stochastic (uncertain), and thus we need a statistic to evaluate the performance of the wind farm. A common statistic is the expected power or the Annual Energy

Production (AEP). Many model simulations are needed to estimate the statistic (Padrón et al., 2016; Murcia et al., 2015). The statistic is usually the objective function of the optimization (Herbert-Acero et al., 2014); thus, the wind farm optimization is an optimization under uncertainty problem (Fig. 1). Optimization under uncertainty (OUU) differs from deterministic optimization in that it contains a nested uncertainty quantification loop to compute statistics. In the OUU problem, for every optimization step many model evaluations are needed to compute the relevant statistics. Thus, even with a very small number

of design variables per turbine, the total number of variables and simulations required by the wind farm optimization can grow very rapidly (Gebraad et al., 2017) and quickly make the problem infeasible, especially when using a high-fidelity model for the wind farm simulation.





We see three approaches to improving wind farm optimization capabilities. Each approach focuses on the different blocks of the optimization under uncertainty problem (Fig. 1c). The first approach is to improve the modeling quality of entire wind farms, i.e., improve the models in all the disciplines (aerodynamics, structures, controls, electrical, acoustics, atmospheric physics, policy, economics, etc.) that are relevant to building and operating a wind farm, as well as the interaction between the different turbines. The second approach is to improve the optimization problem formulation and the algorithms to solve the optimization. And the third approach is to improve the treatment of the stochastic nature of the problem, i.e., develop better uncertainty quantification methods to efficiently compute the relevant wind farm statistics (and their gradients with respect to the design variables of the problem) used in the optimization under uncertainty problem.

The first approach increases the fidelity of the model whereas the second (optimization) and third (uncertainty quantification) approaches seek to reduce the number of model evaluations, as this enables the study of larger and more realistic problems.

Here, we focus on the uncertainty quantification approach (the third approach), as it has not been considered in detail before. The most recent and thorough review of the wind farm optimization literature (Herbert-Acero et al., 2014) does not mention it. It only mentions the first two approaches. In the existing work in the literature, the third approach typically focuses on simple integration methods to compute the statistics, which quantify the effect of the stochastic inputs (Kusiak and Song, 2010; Kwong et al., 2012; Chowdhury et al., 2013; Fleming et al., 2016; Gebraad et al., 2017). Simple integration methods, such as the rectangle rule, are inefficient in the number of samples (simulations of the model) needed to accurately estimate a statistic, such as the AEP. They are especially inefficient if multiple stochastic inputs are considered simultaneously. Normally only the wind direction and/or the wind speed are considered as stochastic input variables. Recent work (Padrón et al., 2016; Murcia et al., 2015) is starting to move beyond these simple integration techniques to compute the AEP and instead using the uncertainty quantification method of polynomial chaos to compute the AEP.

In this paper, which is meant as a comprehensive introduction to uncertainty quantification methods applied to wind-farm simulations, we describe in detail the polynomial chaos (PC) method and show that, for the efficient (small number of model simulations) computation of the AEP, the PC method based on regression should be used. An additional benefit of the PC method is that it makes it feasible to consider multiple uncertain variables (e.g., wind direction, wind speed, wind turbulence, wake model parameter) that impact the computation of the AEP (Padrón, 2017).

In addition, we show how to compute gradients of the statistics, such as the AEP, with a modified version of the PC method. The use of gradients allow us to efficiently tackle much larger optimization problems (Gebraad et al., 2017). To compute the gradients of the wake model, we use the recently developed Floris wake model with the modifications by Thomas et al. (2017) that provide analytic and continuous gradients of the wake model.

We first discuss the details of computing the power and the AEP of a wind farm in Sect. 2. Then, we discuss uncertainty quantification in Sect. 3 and the polynomial chaos method in Sect. 4. Finally, we discuss the details of the problem formulation in Sect. 5 and the results in Sect. 6.



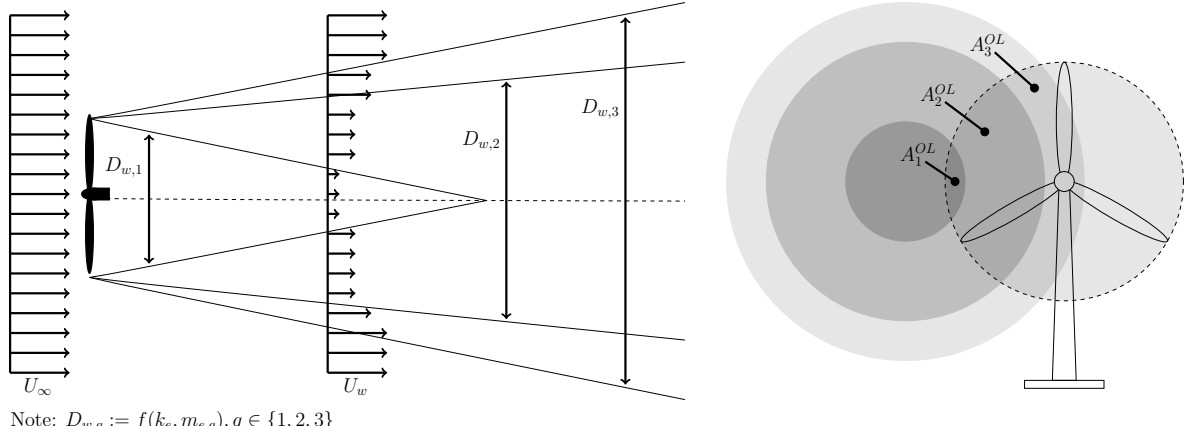

Note: $D_{w,q} := f(k_e, m_{e,q}), q \in \{1, 2, 3\}$

**Figure 2.** Schematic of the Floris wake model. The model has three zones with varying diameters, $D_{w,q}$, that depend on tuning parameters $k_e$ and $m_{e,q}$. The effective hub velocity is computed using the overlap ratio, $A_q^{OL}$, of the part of the rotor-swept area overlapping each wake zone respectively to the total rotor-swept area.

## 2 Computing the power and the annual energy production of a wind farm

We first describe the aerodynamic wake model we use (Sect. 2.1). The wake model gives an estimate of the hub-height velocity at each wind turbine, from which we can compute the power produced by the wind farm (Sect. 2.2). Then, to obtain the Annual Energy Production (AEP) we need to integrate the power over all wind conditions that occur in a year (Sect. 2.3) and weigh the results proportionally to the frequency with which such wind conditions manifest themselves.

### 2.1 Floris

The Floris (FLow Redirection and Induction in Steady-state)[1] (Gebraad et al., 2016) wake model is an enhancement of the Jensen wake model (Jensen, 1983) and the wake deflection model presented in (Jiménez et al., 2009). The Floris model builds on the Jensen model by defining three separate wake zones with differing expansion and decay rates (controlled by tunable coefficients) to more accurately describe the velocity deficit across the wake region. A simple overlap ratio of the area of the rotor in each zone of each shadowing wake to the full rotor area is used to determine the effective hub velocity of a given turbine. A simple overview of the Floris model, showing the zones and overlap areas, is shown in Fig. 2. We use the Floris wake model with changes to remove discontinuities and add curvature to regions of non-physical zero gradient to make the model more suitable for gradient-based optimization (Thomas et al., 2017). In this work, we use the parameter values recommended in (Gebraad et al., 2016) and (Thomas et al., 2017) and set the yaw-offset angle of each turbine to zero.

[1]We use the name Floris for the model, instead of FLORIS, the name used in (Gebraad et al., 2016).





**Table 1.** The variables used for calculating the power.

| | |
|---|---|
| Uncertain $\boldsymbol{\xi}$ | wind direction. |
| | freestream wind speed. |
| Design $\mathbf{x}$ | $x$ - The x location of each turbine. |
| | $y$ - The y location of each turbine. |
| Parameters $\boldsymbol{\theta}$ | yaw angles, turbine characteristics, |
| | and wake model parameters. |

## 2.2 Computing the power of a wind farm

We will consider the power of the wind farm to be a function of three classes of variables: uncertain variables $\boldsymbol{\xi}$, design variables $\mathbf{x}$, and parameters $\boldsymbol{\theta}$,

$$P = P(\boldsymbol{\xi}, \mathbf{x}, \boldsymbol{\theta}). \tag{1}$$

Uncertain variables are variables that follow a probability distribution, design variables are variables that an optimizer can vary, and parameters are important constants that govern the behavior of the system. The classification of the variables is problem dependent. For instance, the rotor yaw could be considered a design variable, a parameter, or an uncertain variable to account for yaw-offset measurement error. A tunable parameter of a wake model, such as the wake expansion coefficient, could be considered as a parameter or as an uncertain variable given by a particular distribution.

For the problems considered in this work, Table 1 lists in which category we place each variable that influences the power computation. The uncertain variables are the wind direction and wind speed with probability distributions described in Sect. 5.1.

The power of the wind farm for a given wind direction and wind speed is equal to the sum of the power produced by each turbine $P = \sum_{i=1}^{n_{turb}} P_i$. The power of each turbine is calculated from

$$P_i = \frac{1}{2}\rho C_P A U_i^3, \tag{2}$$

where $\rho$ is the air density, $A$ is the rotor swept area, $C_P$ is the power coefficient, and $U_i$ is the effective hub velocity for each turbine, which is calculated by the wake model and is a function of the three types of variables described above

$$U_i = f(\boldsymbol{\xi}, \mathbf{x}, \boldsymbol{\theta}). \tag{3}$$

The power coefficient captures both the aerodynamic and electromechanical properties of the wind turbine. It is a complex function of many variables (Herbert-Acero et al., 2014) and is usually reported by wind turbine manufacturers as a function of 20 the tip-speed ratio, which depends on wind speed at hub height $U_i$. A simple expression for $C_p$ can also be computed using the classical actuator disk theory (Sanderse, 2009).





### 2.3 Computing the Annual Energy Production (AEP) of a wind farm

The Annual Energy Production (AEP) is an important metric used to describe a wind farm. The AEP is a statistic. Specifically, it is a mean, as it is a function of the expected power multiplied by the number of hours in a year,

$$AEP = 8760 \frac{hr}{yr} \mathrm{E}[P(\boldsymbol{\xi})]. \tag{4}$$

The expected power, $\mathrm{E}[P(\boldsymbol{\xi})]$, or the mean of the power, $\mu_P$, is defined as

$$\mu_P = \mathrm{E}[P(\boldsymbol{\xi})] = \int_\Omega P(\boldsymbol{\xi})\rho(\boldsymbol{\xi})\,d\boldsymbol{\xi}, \tag{5}$$

where $\boldsymbol{\xi} = (\xi_1, \xi_2, \ldots, \xi_n)$ is a vector of random variables, which we refer to as the uncertain variables, $\rho(\boldsymbol{\xi})$ is the joint probability density function of the uncertain variables, $\Omega$ is the domain of the uncertain variables, and $P$ is the power produced by the wind farm (Eq. (1)). Common uncertain variables are the wind direction and the freestream wind speed.

The expected power, and hence the AEP, is normally computed as a weighted average, which amounts to the rectangle rule of integration (Sect. 3.1.1). Other uncertainty quantification methods (Sect. 3) can be used to compute the expected value (AEP). Specifically, we can compute the AEP efficiently by polynomial chaos (Sect. 4).

### 3 Uncertainty quantification

Uncertainty quantification (UQ) is the process of (1) characterizing input uncertainties, and then (2) propagating these in-
put uncertainties through a computational model with the goal of quantifying their effect on the model's output. There are many sources of uncertainty in the modeling of a problem, and different classifications of the uncertainties have been proposed (Kennedy and O'Hagan, 2001; Oberkampf et al., 2001; Beyer and Sendhoff, 2007). A common classification is to divide the uncertainty into aleatory and epistemic uncertainties (Oberkampf et al., 2001).

In this work, we consider aleatory uncertainties that arise from the variability in the inputs to our model caused by changing
environmental conditions. We describe this input variability as random variables with associated probability distributions. Thus, the first step of characterizing the input uncertainties is concerned with finding the probability distributions that describe the model's inputs. This process is known as statistical inference, model calibration and inverse uncertainty quantification (Smith, 2014). Here, we assume that this step has been completed, i.e., we have distributions that characterize the uncertain inputs (Sect. 5.1). We focus on the second step of propagating the input uncertainties to find the statistics that describe the output.

### 3.1 Uncertainty propagation methods

The goal of uncertainty propagation methods is to compute the statistics that describe the effect of the uncertain inputs on the model output. There are several methods to propagate the uncertainties and compute statistics (Le Maître and Knio, 2010; Smith, 2014), where each method has its advantages/disadvantages depending on the type and size of the problem. The most common methods are sampling or Monte Carlo methods (Caflisch, 1998). Other methods include direct integration methods

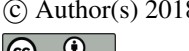



and stochastic expansion methods. Direct integration methods are the currently used method to compute the AEP of a wind farm; we briefly describe them below (Sect. 3.1.1). We describe the stochastic expansion method of polynomial chaos in detail in Sect. 4, and compare the different methods to propagate the uncertainties to compute the AEP in Sect. 6.2.

### 3.1.1 Direct numerical integration (Rectangle rule)

As the name implies, this method numerically evaluates the integrals in the definition of the statistics. The integrals to evaluate for the mean (or expected value) and the variance are

$$\mu_R = \mathrm{E}[R] = \int_\Omega R(\boldsymbol{\xi})\rho(\boldsymbol{\xi})\,d\boldsymbol{\xi}, \tag{6}$$

$$\sigma_R^2 = \mathrm{Var}[R] = \mathrm{E}[(R(\boldsymbol{\xi}) - \mathrm{E}[R(\boldsymbol{\xi})])^2] \tag{7}$$

$$= \mathrm{E}[R(\boldsymbol{\xi})^2] - (\mathrm{E}[R(\boldsymbol{\xi})])^2 \tag{8}$$

$$= \int_\Omega R(\boldsymbol{\xi})^2 \rho(\boldsymbol{\xi})\,d\boldsymbol{\xi} - \mu_R^2, \tag{9}$$

where $R(\boldsymbol{\xi})$ is the model output and $\boldsymbol{\xi}$ the uncertain input variables. The random vector $\boldsymbol{\xi} = (\xi_1, \xi_2, \ldots, \xi_n)$ with joint probability distribution $\rho(\boldsymbol{\xi})$ describes the input variability over the domain $\Omega$. Each random input can follow a particular distribution $\rho_i(\xi_i)$. For the case of independent random variables, the joint distribution is the product of each univariate distribution $\rho(\boldsymbol{\xi}) = \prod_{i=1}^n \rho_i(\xi_i)$. In this work, we will assume that the random input variables are independent. For a description of methods

for dealing with cases when the variables are dependent see Padrón (2017).

There are many quadrature methods to evaluate integrals (Ascher and Greif, 2011). We describe the rectangle rule, as this is what is currently used in the wind farm community to compute the AEP.

**Rectangle rule**. The rectangle rule, or mid-point rule, is the simplest and most straightforward quadrature method. To approximate the mean or expected value,

$$\mu_R = \mathrm{E}[R] = \int_\Omega R(\boldsymbol{\xi})\rho(\boldsymbol{\xi})\,d\boldsymbol{\xi} \tag{6 revisited}$$

with the rectangle rule, we divide the domain of the uncertain variable[2] $\Omega = [a, b]$ into $m$ equal subintervals of length $\Delta\xi = (b-a)/m$. Next, we construct rectangles with base $B = \Delta\xi$ and height equal to the product of the response of the model and the density evaluated at the mid-point of the subinterval $H = R(\xi_j)\rho(\xi_j)$. Then, the rectangle rule approximates the expected value by adding up the areas of the $m$ rectangles

$$\mathrm{E}[R] = \int_a^b R(\xi)\rho(\xi)\,d\xi \approx \sum_{j=1}^m R(\xi_j)\rho(\xi_j)\Delta\xi. \tag{10}$$

---

[2]For simplicity, we consider the 1-dimensional case, $\boldsymbol{\xi} = \xi$.





A simple improvement is to integrate the density exactly within each subinterval

$$\mathrm{E}[R] = \int_a^b R(\xi)\rho(\xi)\,d\xi \approx \sum_{j=1}^m R(\xi_j) \int_{\xi_{j-1/2}}^{\xi_{j+1/2}} \rho(\xi)\,d\xi.$$ (11)

This is easily done as the density is known. This modification is helpful for a small number of evaluations $m$. We will use this modified rectangle rule and simply refer to it as the rectangle rule.

## 4  Polynomial chaos

Polynomial chaos (PC) is the name of an uncertainty quantification (UQ) method that approximates a function with a polynomial expansion made up of orthogonal polynomials. This function has *random variables* as inputs, and we are interested in the effects of the random (uncertain) inputs on the output of this function. Statistics of the output can describe the effects of the inputs. We use the polynomial chaos method to efficiently compute these statistics and the gradients of these statistics.

We first describe the polynomial chaos method in Sect. 4.1. We then discuss two methods—quadrature and regression—to compute the coefficients used in the PC expansion (Sect. 4.2). We finish with an extension of polynomial chaos to compute gradients (Sect. 4.3).

### 4.1  Polynomial chaos expansion

Let $R(\xi)$ be a function of interest that depends on the uncertain variable $\xi$. We can approximate the function by a polynomial expansion

$$R(\xi) \approx \hat{R}(\xi) = \sum_{i=0}^p \alpha_i \phi_i(\xi).$$ (12)

The approximate response $\hat{R}(\xi)$ is a polynomial of order $p$. Usually, the larger the polynomial order the closer the approximation is to the true response $R(\xi)$.

The polynomial basis $\{\phi_i(\xi)\}_{i=0}^p$ is determined by the distribution of the uncertain variable—the polynomial basis is orthogonal with a weight function that corresponds up to a constant to the probability density function of the uncertain variable. Common random (uncertain) variables (Normal, Uniform, Exponential, Beta) have corresponding classical orthogonal polynomials (Hermite, Legendre, Laguerre, Jacobi) (Eldred et al., 2008). It is necessary to numerically generate orthogonal polynomials for uncertain variables with empirically-determined distributions, such as those obtained from wind conditions, to preserve the optimal convergence property of the polynomial chaos expansion (Oladyshkin and Nowak, 2012). Also, the use of orthogonal polynomials allows us to analytically compute statistics from the polynomial chaos expansion (Sect. 4.1.1). Details about the numerical generation of orthogonal polynomials can be found in Gautschi (2004) and an example of the generation of orthogonal polynomials for wind distributions in Padrón (2017). In addition to the orthogonal polynomials, the other component of the expansion Eq. (12) are the coefficients $\alpha_i$. The coefficients can be computed either by quadrature or regression as described in Sect. 4.2.

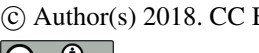



For the case of multiple uncertain variables $\boldsymbol{\xi} = (\xi_1, \xi_2, \ldots, \xi_n)$ and using a multi-index $\mathbf{i} = (i_1, i_2, \ldots, i_n)$, we write the multi-dimensional polynomial approximation as

$$R(\boldsymbol{\xi}) \approx \hat{R}(\boldsymbol{\xi}) = \sum_{\mathbf{i} \in \mathcal{I}_p} \alpha_{\mathbf{i}} \Phi_{\mathbf{i}}(\boldsymbol{\xi}). \tag{13}$$

The multi-dimensional basis functions $\Phi_{\mathbf{i}}(\boldsymbol{\xi})$ are given by products of the 1-dimensional orthogonal polynomials

$$\Phi_{\mathbf{i}}(\boldsymbol{\xi}) = \prod_{j=1}^{n} \phi_{i_j}(\xi_j). \tag{14}$$

When the uncertain variables are independent, the multi-dimensional basis functions are also orthogonal. The values of the elements $i_j$ of the multi-index depend on how the expansion is truncated, i.e., on how the index set $\mathcal{I}_p$ is defined. There are two common ways in which to define the index set: *total-order expansion* and *tensor-product expansion*.

In *total-order expansion* a total polynomial order bound $p$ is enforced:

$$\mathcal{I}_p = \{\mathbf{i} : |\mathbf{i}| \le p\}, \qquad |\mathbf{i}| = i_1 + i_2 + \cdots + i_n. \tag{15}$$

And in *tensor-product expansion* a per-dimension polynomial order bound $p_j$ is enforced

$$\mathcal{I}_p = \{\mathbf{i} : i_j \le p_j, \; j = 1, \ldots, n\}. \tag{16}$$

An example showing the multi-dimensional basis polynomials Eq. (14) for both the *total-order expansion* and *tensor-product expansion* can be found in Padrón (2017). The *tensor-product expansion* is the preferred approach when the coefficients are computed with quadrature (Sect. 4.2.1) because of increased monomial coverage and accuracy (Eldred and Burkardt, 2009).

The *total-order expansion* is the preferred approach when the coefficients are computed with regression (Sect. (4.2.2)) because it keeps the sampling requirements lower (Eldred and Burkardt, 2009).

### 4.1.1 Mean and variance from the polynomial chaos expansion

The mean and variance of the function of interest $R(\boldsymbol{\xi})$ are a function of the coefficients $\alpha_i$ of the polynomial chaos expansion.

The statistics are obtained by substituting the polynomial chaos expansion Eq. (13) into the definitions of the mean Eq. (6) and variance Eq. (9), and by integrating the expansion and simplifying using the orthogonality of the polynomials.

The mean is the zeroth coefficient

$$\mu_R = \alpha_{\mathbf{0}} \tag{17}$$

and the variance is the sum of the product of the square of the coefficients—excluding the zeroth coefficient—with the inner

product $\langle \Phi_{\mathbf{i}}^2(\boldsymbol{\xi}) \rangle$,

$$\sigma_R^2 = \sum_{\mathbf{i} \in \mathcal{I}_p \setminus \{\mathbf{0}\}} \alpha_{\mathbf{i}}^2 \langle \Phi_{\mathbf{i}}^2(\boldsymbol{\xi}) \rangle, \tag{18}$$

where the inner product is defined as $\langle \Phi_{\mathbf{i}}^2(\boldsymbol{\xi}) \rangle = \int_{\Omega} \Phi_{\mathbf{i}}(\boldsymbol{\xi}) \Phi_{\mathbf{i}}(\boldsymbol{\xi}) \rho(\boldsymbol{\xi}) \, d\boldsymbol{\xi}$ and $\mathbf{0}$ is the first multi-index—the one with all zero elements.




## 4.2 Calculating polynomial chaos coefficients

The coefficients of the polynomial chaos expansion Eq. (13) can be calculated via quadrature or by linear regression.

### 4.2.1 Quadrature

To obtain the coefficients of the polynomial chaos expansion

$$R(\boldsymbol{\xi}) = \sum_{\mathbf{i} \in \mathcal{I}_p} \alpha_{\mathbf{i}} \Phi_{\mathbf{i}}(\boldsymbol{\xi}), \tag{19}$$

via quadrature, we take the inner product of both sides of Eq. (19) with respect to $\Phi_{\mathbf{j}}(\boldsymbol{\xi})$ to yield

$$\langle R, \Phi_{\mathbf{j}} \rangle = \sum_{\mathbf{i} \in \mathcal{I}_p} \alpha_{\mathbf{i}} \langle \Phi_{\mathbf{i}}, \Phi_{\mathbf{j}} \rangle. \tag{20}$$

Making use of the orthogonality of the polynomials and solving for the coefficients in Eq. (20), we obtain

$$\alpha_{\mathbf{i}} = \frac{\langle R(\boldsymbol{\xi}), \Phi_{\mathbf{i}}(\boldsymbol{\xi}) \rangle}{\langle \Phi_{\mathbf{i}}^2(\boldsymbol{\xi}) \rangle} = \frac{1}{\langle \Phi_{\mathbf{i}}^2(\boldsymbol{\xi}) \rangle} \int\limits_{\Omega} R(\boldsymbol{\xi}) \Phi_{\mathbf{i}}(\boldsymbol{\xi}) \rho(\boldsymbol{\xi}) \, d\boldsymbol{\xi}, \tag{21}$$

where the domain $\Omega$ is the Cartesian product of $1D$ domains $\Omega_j$ for each dimension, $\Omega = \Omega_1 \times \cdots \times \Omega_n$, and $\rho(\boldsymbol{\xi}) = \prod_{j=1}^{n} \rho_j(\xi_j)$ is the joint probability density of the stochastic parameters. The inner product $\langle \Phi_{\mathbf{i}}^2(\boldsymbol{\xi}) \rangle$ is known analytically or inexpensively computed. Thus, most of the computational expense in solving for the coefficients resides in evaluating the model $R(\boldsymbol{\xi})$ in the multi-dimensional integral $\int_{\Omega} R(\boldsymbol{\xi}) \Phi_{\mathbf{i}}(\boldsymbol{\xi}) \rho(\boldsymbol{\xi}) \, d\boldsymbol{\xi}$. This integral is solved with quadrature (numerical integration). Note that the zero coefficient in Eq. (21) reduces to the definition of the mean

$$\mu_R = \alpha_0 = \int\limits_{\Omega} R(\boldsymbol{\xi}) \rho(\boldsymbol{\xi}) \, d\boldsymbol{\xi}, \tag{22}$$

which the direct numerical integration methods attempt to compute directly (Sect. 3.1.1).

### 4.2.2 Regression

To obtain the coefficients of the polynomial chaos expansion Eq. (13) via regression, we construct a linear system

$$\boldsymbol{\Phi}\boldsymbol{\alpha} = R \tag{23}$$

and solve for the coefficients $\boldsymbol{\alpha}$ that best represent a set of responses $R$. The set of responses is generated by evaluating the model at $m$ realizations of the uncertain vector $\boldsymbol{\xi}$. The $m$ uncertain vectors are most commonly obtained by sampling the density of the uncertain variables (Hosder et al., 2007).

Each row of the matrix $\boldsymbol{\Phi}$ contains the orthogonal polynomials $\Phi_{\mathbf{j}}$ evaluated at a sample $\boldsymbol{\xi}_i$

$$\begin{bmatrix} \Phi_{\mathbf{0}}(\boldsymbol{\xi}_1) & \cdots & \Phi_{\mathbf{n-1}}(\boldsymbol{\xi}_1) \\ \vdots & \ddots & \vdots \\ \Phi_{\mathbf{0}}(\boldsymbol{\xi}_m) & \cdots & \Phi_{\mathbf{n-1}}(\boldsymbol{\xi}_m) \end{bmatrix} \begin{bmatrix} \alpha_0 \\ \vdots \\ \alpha_{n-1} \end{bmatrix} = \begin{bmatrix} R_1 \\ \vdots \\ R_m \end{bmatrix}. \tag{24}$$





The size of the $m \times n$ matrix is determined by the number of samples $m$ and by how the polynomial chaos expansion is truncated (Sect. 4.1) which results in $n$ terms. It is common to specify a total order expansion along with a collocation ratio $cr = m/n$ to determine the number of samples $m$. The collocation ratio determines if the system is overdetermined $cr > 1$ or underdetermined $cr < 1$.

5      For overdetermined systems, the most popular method (and the one we use) to estimate the coefficients is *least squares*, in which we pick coefficients $\boldsymbol{\alpha} = (\alpha_0, \alpha_1, \ldots, \alpha_{n-1})$ that minimize the residual sum of squares

$$\boldsymbol{\alpha} = \arg \min ||\boldsymbol{\Phi}\boldsymbol{\alpha} - R||_2^2. \tag{25}$$

For underdetermined systems, solving a regularized least squares problem is preferred (Doostan and Owhadi, 2011).

     For a given number of samples $m$ in the linear system Eq. (24) we can use cross-validation (Hastie et al., 2009) to pick the 10   best polynomial order $n$ to approximate the response.

### 4.3   Gradients of statistics with polynomial chaos

Let $R(\boldsymbol{\xi}, \mathbf{x})$ be a function of interest that depends on uncertain variables $\boldsymbol{\xi}$ and also on design variables $\mathbf{x}$. We assume independence between the design and uncertain variables[3]. Now the polynomial chaos expansion—over the uncertain variables—becomes

$$R(\boldsymbol{\xi}, \mathbf{x}) \approx \hat{R}(\boldsymbol{\xi}, \mathbf{x}) = \sum_{\mathbf{i} \in \mathcal{I}_p} \alpha_{\mathbf{i}}(\mathbf{x}) \boldsymbol{\Phi}_{\mathbf{i}}(\boldsymbol{\xi}). \tag{26}$$

This expansion is only valid for a particular design vector—the coefficients $\alpha_{\mathbf{i}}(\mathbf{x})$ are a function of the design variables. Therefore, the statistics are also a function of the design variables. Specifically the mean and the variance are

$$\mu_R(\mathbf{x}) = \alpha_{\mathbf{0}}(\mathbf{x}), \tag{27}$$

$$\sigma_R^2(\mathbf{x}) = \sum_{\mathbf{i} \in \mathcal{I}_p \backslash \mathbf{0}} \alpha_{\mathbf{i}}^2(\mathbf{x}) \langle \boldsymbol{\Phi}_{\mathbf{i}}^2(\boldsymbol{\xi}) \rangle. \tag{28}$$

20   #### 4.3.1   Gradients of the statistics with polynomial chaos

We want to know the gradients of the statistics with respect to the design variables, and we proceed to derive them below. For simplicity, we drop the subscript from the statistics $\mu_R = \mu$, the explicit variable dependence $R(\boldsymbol{\xi}, \mathbf{x}) = R$, the bolded notation, and we use the following notation for the gradient $\frac{df}{dx} \equiv \nabla f$.

     The gradient of the mean from Eq. (27) is

$$\frac{d\mu}{dx} = \frac{d\alpha_{\mathbf{0}}}{dx}, \tag{29}$$

and the gradient of the variance from Eq. (28) is

$$\frac{d\sigma^2}{dx} = \sum_{\mathbf{i} \in \mathcal{I}_p \backslash \mathbf{0}} \langle \boldsymbol{\Phi}_{\mathbf{i}}^2 \rangle \frac{d\alpha_{\mathbf{i}}^2}{dx} = 2 \sum_{\mathbf{i} \in \mathcal{I}_p \backslash \mathbf{0}} \langle \boldsymbol{\Phi}_{\mathbf{i}}^2 \rangle \alpha_{\mathbf{i}} \frac{d\alpha_{\mathbf{i}}}{dx}. \tag{30}$$

---

[3]For most applications, the design and uncertain variables are independent. For instance, the design variables are the wind turbines location and the uncertain variables are the wind conditions.



Both, the mean and the variance gradients, depend on the gradient of the coefficients $\frac{d\alpha_{\mathbf{i}}}{dx}$.

### 4.3.2 Gradients of the coefficients

The gradient of the coefficients can be computed with quadrature or regression, similarly to how the coefficients can be calculated with quadrature (Sect. 4.2.1) or regression (Sect. 4.2.2).

5    **Quadrature**. We start from the equation for the coefficients Eq. (21) and take the gradient to obtain

$$\frac{d\alpha_{\mathbf{i}}}{dx} = \frac{1}{\langle \Phi_{\mathbf{i}}^2 \rangle} \int\limits_{\Omega} \frac{dR}{dx} \Phi_{\mathbf{i}} \rho \, d\boldsymbol{\xi} = \frac{\langle \frac{dR}{dx}, \Phi_{\mathbf{i}} \rangle}{\langle \Phi_{\mathbf{i}}^2 \rangle}. \tag{31}$$

Replacing this equation into the gradient of the mean Eq. (29) we obtain

$$\frac{d\mu}{dx} = \left\langle \frac{dR}{dx} \right\rangle. \tag{32}$$

And replacing Eq. (31) into the gradient of the variance Eq. (30) we obtain

$$\frac{d\sigma^2}{dx} = 2 \sum_{\mathbf{i} \in \mathcal{I}_p \backslash \mathbf{0}} \alpha_{\mathbf{i}} \left\langle \frac{dR}{dx}, \Phi_{\mathbf{i}} \right\rangle. \tag{33}$$

To obtain the gradients of the statistics with respect to each design variable we need to evaluate the multi-dimensional integral containing $\frac{dR}{dx}$. The integral is evaluated with quadrature (Sect. 4.2.1) and requires the computation of the gradient of the response at each of the quadrature points. Ideally, one would use adjoint methods Giles and Pierce (2000) or algorithmic differentiation Griewank and Walther (2008) to compute the gradients, $\frac{dR}{dx}$, efficiently.

15    **Regression**. We start from the linear system Eq. (23) and take the gradient to obtain

$$\frac{d\boldsymbol{\Phi}\boldsymbol{\alpha}}{dx} = \frac{dR}{dx} \tag{34}$$

$$\boldsymbol{\Phi}\frac{d\boldsymbol{\alpha}}{dx} = \frac{dR}{dx} \tag{35}$$

$$\begin{bmatrix} \Phi_{\mathbf{0}}(\boldsymbol{\xi}_1) & \cdots & \Phi_{\mathbf{n-1}}(\boldsymbol{\xi}_1) \\ \vdots & \ddots & \vdots \\ \Phi_{\mathbf{0}}(\boldsymbol{\xi}_m) & \cdots & \Phi_{\mathbf{n-1}}(\boldsymbol{\xi}_m) \end{bmatrix} \begin{bmatrix} \frac{d\alpha_0}{dx_1} & \cdots & \frac{d\alpha_0}{dx_d} \\ \vdots & \ddots & \vdots \\ \frac{d\alpha_{n-1}}{dx_1} & \cdots & \frac{d\alpha_{n-1}}{dx_d} \end{bmatrix} = \begin{bmatrix} \frac{dR_1}{dx_1} & \cdots & \frac{dR_1}{dx_d} \\ \vdots & \ddots & \vdots \\ \frac{dR_m}{dx_1} & \cdots & \frac{dR_m}{dx_d} \end{bmatrix}. \tag{36}$$

To solve for the gradient of the coefficients, we solve the linear system one column at a time of the $\frac{d\boldsymbol{\alpha}}{dx}$ matrix with the 20   corresponding column of the matrix of the gradients $\frac{dR}{dx}$. The linear system for the multiple right hand sides can be solved with the methods described in Sect. 4.2.2.

Again, the gradient of the mean Eq. (29) and the gradient of the variance Eq. (30) are a function of the gradient of the coefficients. Thus, to obtain the gradient of the mean take the first row of the $\frac{d\boldsymbol{\alpha}}{dx}$ matrix; and for the gradient of the variance use the gradient of the coefficients from all the other rows.





### 4.3.3 Gradients of the statistics by direct numerical integration

Similarly to computing the mean and variance with direct numerical integration (Sect. 3.1.1), we can also compute the gradients of the mean and variance directly with numerical integration by differentiating the definitions of the mean, Eq. (6), and the variance, Eq. (9).

## 5 Problem details

Here we describe the details of computing AEP in the wind farm. We first describe and discuss the inputs, which are the wind direction and wind speed (uncertain variables) (Sect. 5.1) and the wind turbine positions (design variables) (Sect. 5.2). Then, we describe details about the output, the average AEP error (Sect. 5.3), that we use to compare the different methods we consider to compute the AEP (Sect. 5.4).

### 5.1 Probability distributions of the uncertain wind conditions

We consider the wind direction and the wind speed as uncertain variables with probability distributions shown in Fig. 3. The distributions show the likelihood of a particular wind direction or wind speed occurring during a year. The wind direction distribution (Fig. 3a) is constructed by linearly interpolating wind measurements taken by the NoordZeeWind meteorological mast during a year (Brand et al., 2012). The mast is situated near the Princess Amalia wind farm (Sect. 5.2), and the measurements were taken from July 1, 2005, to June 30, 2006, before the construction of the nearby Noordzee offshore wind farm. The zero degree direction is set at North and increases clockwise.

The wind speed is usually correlated with the wind direction, but for simplicity, we assume that they are independent (Sect. 3.1.1). With this assumption, we fit a single Weibull distribution to the meteorological mast data (Fig. 3b). The Weibull distribution is the preferred distribution to model the wind speed distribution (Belu and Koracin, 2013; Herbert-Acero et al., 2014). For the Princess Amalia data, the resulting wind speed distribution is the Weibull

$$\rho(\xi; \alpha, \beta) = \frac{\alpha}{\beta} \left( \frac{\xi}{\beta} \right)^{\alpha-1} e^{-(\xi/\beta)^{\alpha}}, \qquad \xi \geq 0, \tag{37}$$

with shape parameter $\alpha = 1.8$ and scale parameter $\beta = 12.552983$. The operational range of the turbine is from the cut-in speed of 3 m/s to the cut-out speed of 25 m/s. We will consider the upper limit to be 20 m/s as, above this speed, the power produced by the wind farms is almost always constant at its rated power and thus contributes only a constant term to AEP.

### 5.2 The wind farm layouts

To showcase the results, we will focus on four representative layouts: Grid, Amalia, Optimized and Random (Fig. 4). The layouts have sixty turbines represented by individual dots in each of the figures we show in the results. Each dot is to scale, where the diameter represents the rotor swept area. The Grid layout fits in a box of equivalent area to that of the Amalia layout. The Amalia layout, which is grid-like, is that of the Princess Amalia wind farm located 23 km off the coast of the Netherlands.

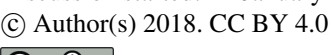



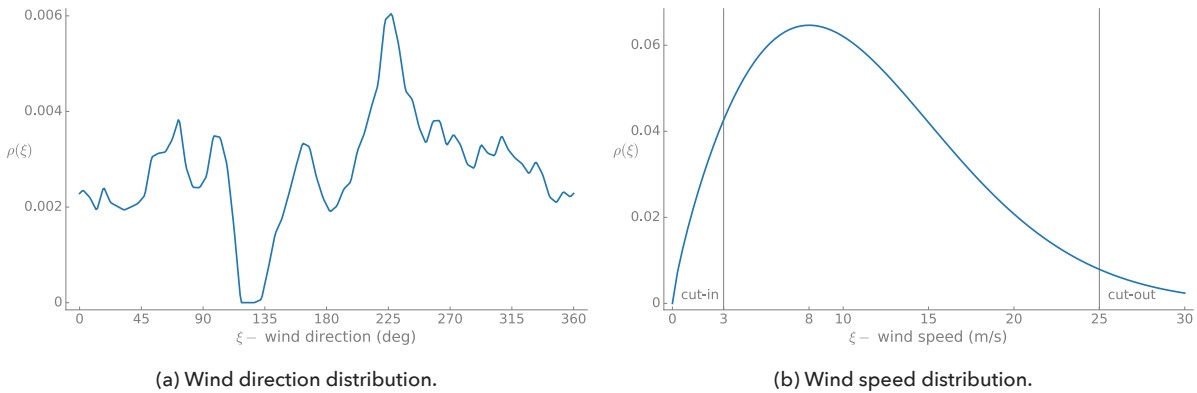

(a) Wind direction distribution.  (b) Wind speed distribution.

**Figure 3.** The uncertain variables probability distributions. Both distributions are constructed from wind measurements taken by the NoordZeeWind meteorological mast during a year. The wind direction distribution is a linear interpolation of the data, and the wind speed distribution is a Weibull fit to the meteorological wind speed data averaged over all wind directions. The vertical lines show the cut-in and cut-out speed of a single wind turbine.

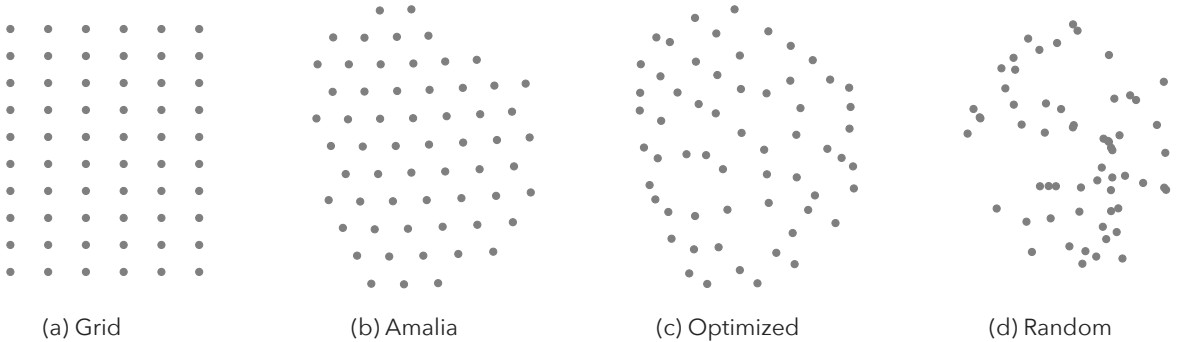

(a) Grid  (b) Amalia  (c) Optimized  (d) Random

**Figure 4.** Representative wind farm layouts used in the results. Each dot represents a wind turbine to scale—the diameter of the dot represents the swept area of the rotor.

The Optimized layout is a representative optimal layout obtained by running the optimization problem (Sect. 6.3.1). When we refer to this particular optimized layout, we will capitalize the word optimized. The Random layout was generated by random sampling and keeping the turbines that are contained within the convex hull of the Amalia wind farm without enforcing any spacing constraints between the turbines. We will refer to the Grid and Amalia as grid-like layouts and the Optimized and Random as non-grid-like layouts.

In reality, the turbines in the Princess Amalia wind farm are the Vestas V80 model. For each of the layouts in our study, we use the NREL 5-MW reference turbine (Jonkman et al., 2009), as this turbine is similar the the Vestas V80, and has an open source design.





### 5.3 Convergence metric — the average AEP error

We use an ensemble of 10 AEP results to compute the average AEP error. The average AEP error allows us to better illustrate the differences between the different methods used to compute the AEP and to avoid drawing conclusions from one-off solutions. We found that averaging over 10 AEP results is enough to illustrate the difference between methods and to smooth

out the convergence of the AEP error (Fig. 8). The ensemble of results account for the fact that the zero (starting) position for the wind direction is arbitrary (it could be North, South, East, West, etc.), and by averaging the AEP errors, it smooths out the AEP convergence curves. We generate the ensemble of AEP results by selecting 10 different input sets. For example, for the rectangle rule, if we consider 36 wind directions the 10 sets are $\{[0, 10, 20, \ldots, 340, 350\,\text{degrees}]; [1, 11, 21, \ldots, 341, 351]; \ldots;$ $[9, 19, 29, 349, 359]\}$. For the polynomial chaos based on quadrature, the quadrature points are the numerically generated

Gaussian quadrature points for the interval. Thus, to create 10 different sets, we pick 10 different intervals, i.e., we pick different starting positions. When considering 36 wind directions the chosen intervals are $\{[0, 360]; [10, 370]; [20, 380]; \ldots;$ $[340, 700]; [350, 710]\}$. For both the rectangle and polynomial chaos based on quadrature, the wind speed points for each set are the same. For the polynomial chaos based on regression and for Monte Carlo, the wind directions and wind speed pairs are generated by sampling the distribution. Thus, to obtain 10 different sets, 10 different samplings are performed. We use the

average AEP error as the convergence metric

$$average\,AEP\,error = \frac{1}{10} \sum_{i=1}^{10} \left| \frac{AEP_i - baseline\,AEP}{baseline\,AEP} \right| \times 100\%. \tag{38}$$

### 5.3.1 Baseline AEP

We take as the baseline or true AEP the AEP computed with 200,000 Monte Carlo samples. We picked 200,000 MC samples to ensure the 99 % confidence interval for the true AEP was smaller than 1 % of the computed AEP value for all layouts.

We consider an AEP within 1 % of the baseline AEP to be accurate, and we will use it as a reference for the results. AEP predictions of real wind farms usually have an error of 10–20 % (Barthelmie et al., 2007; Briggs, 2013) due to uncertainty in wind conditions and to the errors of wake models, thus resolving the AEP to less than 1 % is unnecessary.

### 5.4 Methods to compute the AEP

Here, we provide the details of the methods used to compute the AEP, as well as, the abbreviations for the methods:

|      |                                       |                   |
|------|---------------------------------------|-------------------|
| rect | rectangle rule                        | (Sect. 3.1.1)     |
| PC-Q | polynomial chaos based on quadrature  | (Sect. 4.2.1)     |
| PC-R | polynomial chaos based on regression  | (Sect. 4.2.2)     |
| MC   | Monte Carlo                           | (Caflisch, 1998)  |

For the quadrature-based methods (rect, PC-Q), we use tensor product quadrature to compute multi-dimensional integrals. We use the same number of points for each dimension because we did not see any benefit in favoring a particular dimension. For PC-Q, we use Gaussian quadrature. For Monte Carlo, we use the traditional Monte Carlo method, i.e., random samples. For



PC-R, we use Latin-Hypercube sampling (McKay et al., 1979) to generate the samples needed to construct the linear system. We solve the linear system with least-squares. For a given number of samples, given by the total polynomial order $p$, we use 10-fold cross-validation to find the least-squares best fit from polynomials of total order 1 up to total order $p$. The sampling methods and the polynomial chaos methods we use are implemented in the open-source DAKOTA toolkit (Adams et al., 2017).

## 6  Results

The Annual Energy Production (AEP) of a wind farm is obtained by integrating the power output of the farm over the different wind conditions. Thus, we first characterize the power output of the wake model for different input conditions (Sect. 6.1). Next, we focus on the convergence of the AEP (Sect. 6.2), and then on the wind farm layout optimization problem to maximize the AEP (Sect. 6.3).

### 6.1  Power response as a function of the uncertain variables

The power production, computed with the Floris wake model, for the four wind farm layouts (Sect. 5.2) as a function of both the wind direction and the wind speed is shown in Fig. 5. The wind direction is measured from North and increases clockwise. The peaks in the contour lines identify wind directions for which there is a poor performance (low power). The grid-like layouts (top row of Fig. 5) have larger peaks due to wind turbines being aligned along particular directions and thus experiencing full-wake conditions. The worst wind direction for the Grid layout is directly from North ($0°$) or South ($180°$) when rows of ten turbines are aligned. The power for the Optimized layout is worst at around $125°$. This shows that the optimizer took into account that the likelihood of the wind coming from the $125°$ direction is minimal (Fig. 3a). For all layouts, as the speed increases, the power increases until the wind farm reaches its rated power of 300 MW.

### 6.2  AEP convergence: polynomial chaos vs. rectangle rule

We consider the AEP as a function of two uncertain variables: the wind speed and the wind direction. The AEP is usually considered a function of these two variables. We compare the convergence of the AEP for the different methods to compute the AEP: polynomial chaos (based on quadrature and regression), the rectangle rule and Monte Carlo (Fig. 6). The polynomial chaos based on regression (PC-R) performs the best for all layouts. It is followed, by the polynomial chaos based on quadrature (PC-Q) and the rectangle rule, which perform similarly. The worst performer is Monte Carlo. The slow convergence of statistics with Monte Carlo is well known. Monte Carlo will start to outperform the other methods when the AEP is a function of a large number (5–10) of uncertain variables, as it does not suffer from the *curse of dimensionality*.

The superior performance of the polynomial chaos based on regression, especially for the grid like-layouts (Grid and Amalia), is due to the following: the polynomial chaos fit based on regression does not chase all the high-frequency oscillations in the power response (Fig. 5). The PC-R fit is usually not higher than an eight total-order polynomial (Sect. 4.1). Whereas, the PC-Q order fit is higher, as it is directly proportional to the number of samples per dimension[4]. A downside of

---

[4]For the two-dimensional problem at 625 samples ($25 \times 25$ grid) the polynomial order in each dimension is 24.





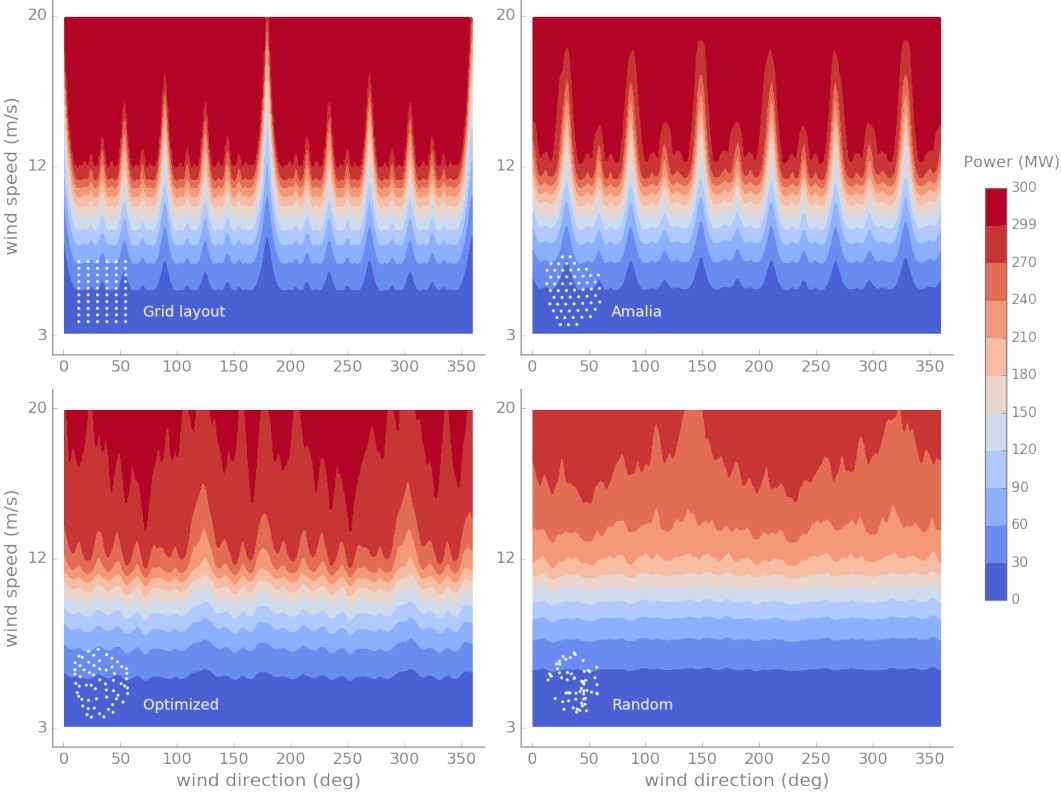

**Figure 5.** Power contours as a function of wind direction and wind speed for the Grid, Amalia, Optimized and Random layouts. The grid-like layouts (top row) have larger peaks due to wind turbines being aligned along particular directions and thus experiencing full-wake conditions. For all layouts as the speed increases the power increases until the wind farm reaches its rated power 300 MW.

the PC-R being able to predict the mean (AEP) accurately, is that it can underpredict the true variance (standard deviation) of the response (Fig. 7). Usually, the standard deviation of the power response (energy) over a year is not considered as a function to optimize. A common objective in wind farm optimization is to maximize the total amount of energy produced over a year independent of the variability in the power production over the year. For a wind farm, the variability of the energy produced over a year is less important than the variability caused by the changing wind conditions during the day.

5      In what follows we will compare the PC-R (the best performing method) with the rectangle rule (the method currently used in practice) to quantify the reduction of samples needed to compute the AEP accurately. Also, we will sometimes use polynomial chaos to refer to polynomial chaos based on regression. Figure 8 only keeps PC-R and the rectangle rule results from Fig. 6, and in addition, the figure shows the average AEP error computed with 10 and 100 sets of samples for each method. For the rectangle rule, there is hardly any difference between the average AEP error computed with 10 or 100 sets.

10    For the PC-R method, the average error with 100 sets shows a smoother convergence. In general, averaging the AEP error over 10 sets of samples is enough to clearly see the differences between the methods used for computing the AEP (Sect. 5.3).





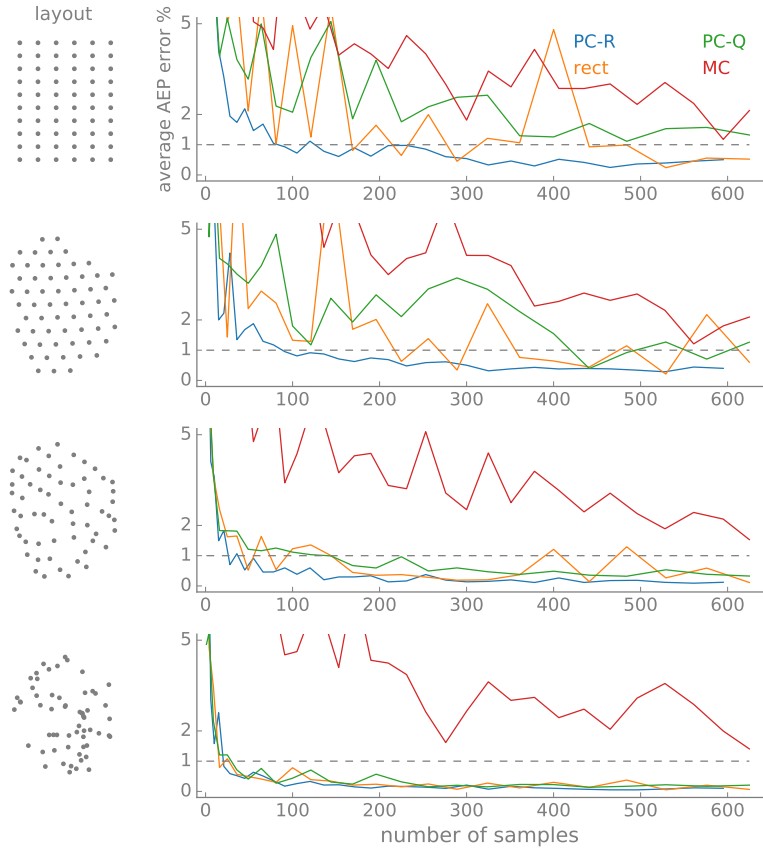

**Figure 6.** The average AEP error as a function of the number of samples for polynomial chaos (based on regression and quadrature), the rectangle rule and Monte Carlo. The AEP is a function of two uncertain variables the wind direction and wind speed. The polynomial chaos based on regression performs best for all layouts.

In Fig. 8, we see that the PC-R convergence curve is consistently below the rectangle rule curve, i.e., PC-R has a smaller error for the same number of samples or the same error for a smaller number of samples. Using the 1 % average AEP error as a metric, we see that PC-R achieves this error with fewer samples than the rectangle rule. The reduction in the number of samples is on the order of six times for the Grid layout, eight times for the Amalia, ten times for the Optimized and no significant improvement for the Random. These reductions in the number of samples are considerable. In addition to providing faster convergence of the AEP, the polynomial chaos based on regression method converges the AEP more smoothly—less oscillatory, more monotone convergence. Smooth convergence is always a desired property, and it especially useful when performing an optimization. Both methods, PC-R and the rectangle rule, perform better for the less grid-like layouts because there is less variability in the power responses as a function of wind direction (see Fig. 5).




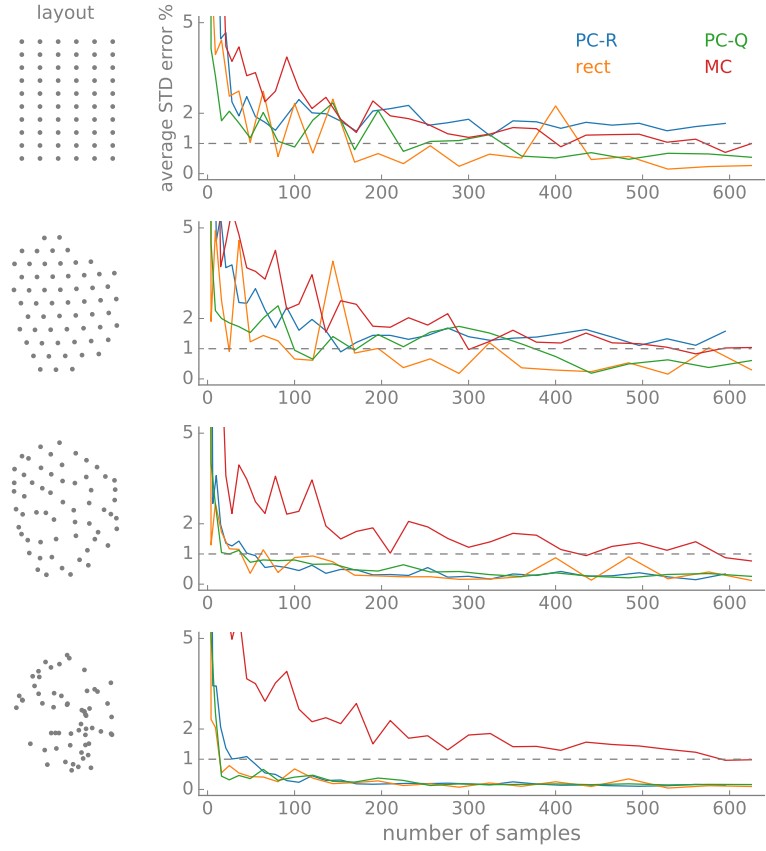

**Figure 7.** The average standard deviation of the energy (STD) error as a function of the number of samples for polynomial chaos (based on regression and quadrature), the rectangle rule and Monte Carlo. The STD is a function of two uncertain variables the wind direction and wind speed. Note that the PC-R response is biased for the Grid and Amalia layouts. The PC-R underpredicts the true STD at the expense of computing the mean (AEP) more accurately (see Fig. 6).

The polynomial chaos response is not only better on average, as we have seen in Fig. 8, but it is also better in general, as shown in Fig. 9. The shaded area in Fig. 9 shows the spread between the maximum and minimum AEP for 10 realizations of each of the number of samples (Sect. 5.3). The solid line shows the average of those 10 realizations and the dashed lines the ±1 % of the baseline AEP. We see that the spread is significantly smaller for the polynomial chaos based on regression and that by around 300 samples the predictions are almost always within 1 % of the true AEP for the grid-like layouts (Grid and Amalia) and around 100 samples for the non-grid-like layouts (Optimized and Random). In contrast, for the rectangle rule, the error in the AEP is still larger than 1 % at 600 samples for the Amalia layout and 500 samples for the Optimized layout.





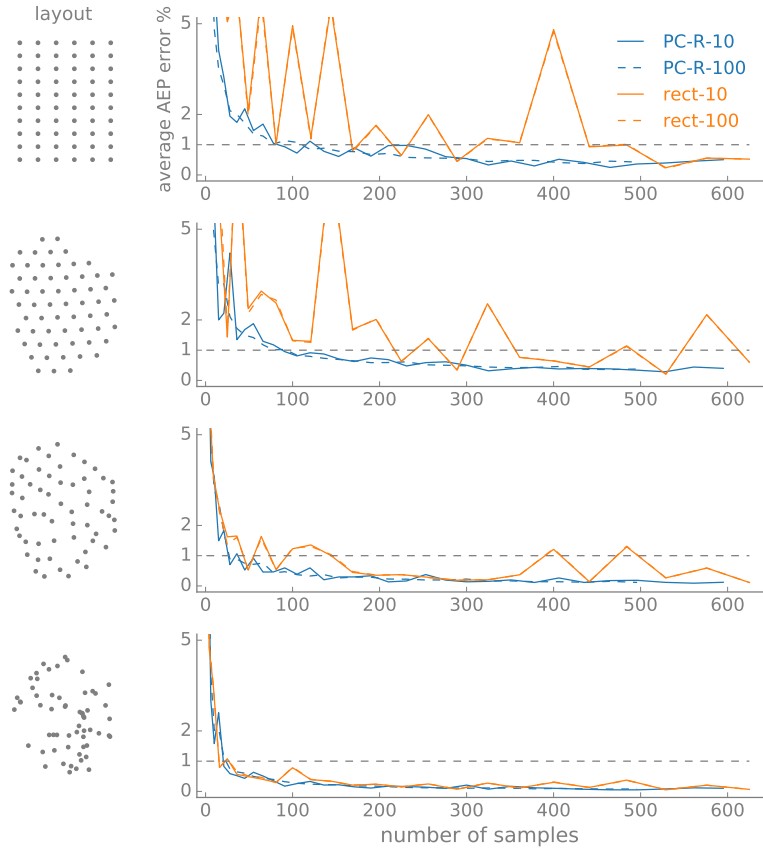

**Figure 8.** The average AEP error as a function of the number of samples for both the rectangle rule and polynomial chaos based on regression (this is the same as Fig. 6, with only keeping the PC-R and the rectangle rule results). In addition, we show the average AEP error computed with 10 (solid line) and 100 (dashed line) sets of samples (Sect. 5.3). The AEP is a function of two uncertain variables the wind direction and wind speed. The polynomial chaos method computes the AEP more accurately with fewer samples.

## 6.3 Wind farm layout optimization

### 6.3.1 Optimization problem

The objective of the wind farm layout optimization is to maximize the AEP (Sect. 2.3) by changing the position of the wind turbines. We assume a fixed number of turbines, 60, of the same type (NREL 5-MW (Jonkman et al., 2009)) and constrain the turbines to stay within a given area and with a minimum separation between them. This objective and constraints result in a





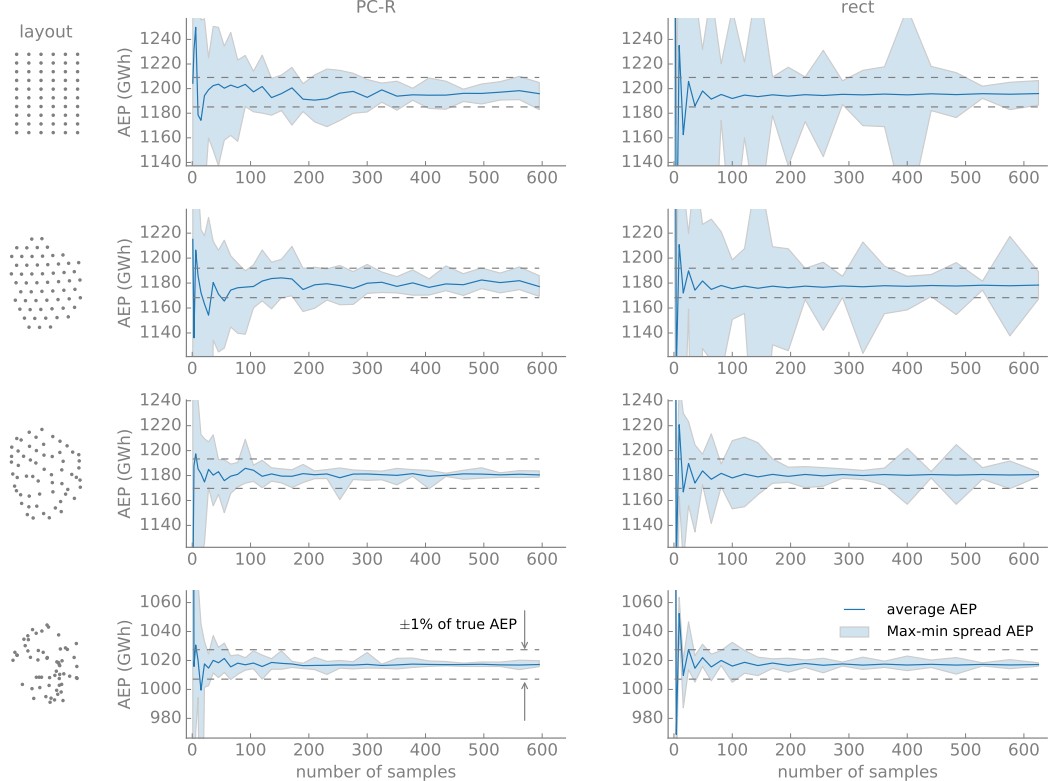

**Figure 9.** Variability in the convergence of the AEP. The AEP is a function of the wind direction and wind speed. The figures of the left column are for polynomial chaos based on regression and those of the right column for the rectangle rule. The shaded area shows the spread between the maximum and minimum AEP for 10 realizations of each of the number of samples. The range in the plots corresponds to $\pm 5$ % of the true AEP for each layout. The variability is significantly smaller for polynomial chaos, which shows that in general, it outperforms the rectangle rule.

nonlinear optimization under uncertainty problem with deterministic constraints:

$$
\begin{aligned}
\underset{x,y}{\text{maximize}} \quad & AEP(x,y,\boldsymbol{\xi}) \\
\text{subject to} \quad & S_{i,j} \geq 2D \ \ i,j = 1\ldots 60 \ \ i \neq j \\
& N_{i,b} \geq 0 \ \ i = 1\ldots 60 \ \ b = 1\ldots 14,
\end{aligned}
\tag{39}
$$

where $S_{i,j}$ is the distance between each pair of turbines $i$ and $j$, and $D$ is the turbine diameter. The normal distance, $N_{i,b}$, from each turbine $i$ to each boundary $b$ is defined as positive when a turbine is inside the boundary, and negative when it is outside of

5  the boundary. The boundary is the convex hull of the Princess Amalia layout—a fourteen-sided convex polygon (dashed-line boundary in the upper-left of Fig. 10). The design variables are the $x$, $y$ coordinates of the 60 wind turbines (same number as



in the Amalia layout), resulting in 120 design variables. The uncertain variables $\boldsymbol{\xi}$ in the objective are the wind direction wind speed.

We solve the optimization problem with the gradient-based sequential quadratic programming optimizer SNOPT (Gill et al., 2005). We use OpenMDAO (Gray et al., 2010) and its wrapper for pyOptSparse (Perez et al., 2012) to call SNOPT from
Python. We scale the variables, constraints and the objective to make them of order one, and set the tolerances to $1 \times 10^{-4}$ per the function (AEP) precision.

### 6.3.2   Optimization results

The AEP in the optimization objective is a function of two uncertain variables the wind direction and wind speed, with probability distributions given in Sect. 5.1. We solve the optimization under uncertainty problem, Eq. (39), for two different starting
layouts: Amalia and Random. We compute the AEP (objective) with different precision (number of samples) and with different methods: the rectangle rule, and polynomial chaos based on quadrature and regression. For each method, we run 10 optimizations, where each optimization uses different sample points to compute the AEP (see Sect. 5.3). The 10 optimizations enable us to get a better understanding of which method is better at finding layouts with high AEP and to avoid drawing conclusions from one-off local optima.

The results of the optimizations are reported in Table 2. The table contains the statistics of the AEP of the optimal layouts for the 10 optimizations of each method. The values of the AEP reported in Table 2 are computed with 200,000 Monte Carlo samples. The same 200,000 samples—wind direction, wind speed pairs—are used to test the optimal layouts obtained by each method.

Polynomial chaos based on regression on average produces the best optimums when the different methods to compute the
AEP use roughly the same number of samples ($\sim 225$). Furthermore, PC-R optimums are comparable with the optimums obtained with the rectangle rule that used 625 samples to compute the AEP.

Starting the optimizations from an already good layout (Amalia) will, in general, find layouts that are better than starting from a bad layout (Random). When considering all the methods, on average, the optimal layouts starting from the Amalia have an AEP that is 0.62 % higher than those starting from the Random layout. However, starting from random layouts gives the
turbines more freedom to move around the design space with the potential of finding novel and better layouts.

The optimal layouts with the maximum AEP for each method and starting layout are shown in Fig. 10. For the optimums obtained starting from the Amalia layout, we see that the distribution of the turbines is similar to the one of the starting layout. And for the optimums starting from the Random layout, we see that the optimal layouts obtained are less grid-like and produce more novel layouts. For each method, as shown in Fig. 10, the optimal layouts starting from the Amalia have a higher AEP
than those from the Random. But a random start has the potential to find better optimums as the turbines explore more of the design space (Fig. 11).

A typical optimization history as a number function calls is shown in Fig. 12. A function call is the AEP computation which requires hundreds (the number of samples specified for each method in Table 2) of calls to the wake model for the wind farm resulting in tens of thousands of calls to the wind farm wake model per optimization. If the gradients of the AEP are computed



**Table 2.** The AEP statistics of the optimized layouts. We generate the statistics for each method from a set of 10 different optimizations. Polynomial chaos based on regression on average produces the best optimums when the different methods to compute the AEP use roughly the same number of samples (∼ 225). Furthermore, PC-R optimums are comparable with the optimums obtained with the rectangle rule that used 625 samples to compute the AEP. Starting the optimizations from an already good layout (Amalia), will in general, find layouts that are better than starting from a bad layout (Random). The values of the AEP reported are computed with 200,000 Monte Carlo samples.

| Method | # samples | Starting layout: Amalia AEP statistics (GWh) | | | | Starting layout: Random AEP statistics (GWh) | | | |
|--------|-----------|------|-----|-----|-----|------|-----|-----|-----|
| | | mean | max | min | std | mean | max | min | std |
| PC-Q | 225 | 1174 | 1179 | 1168 | 3 | 1163 | 1169 | 1151 | 5 |
| PC-R | 231 | 1182 | 1184 | 1179 | 1 | 1178 | 1183 | 1173 | 3 |
| rect | 225 | 1179 | 1180 | 1177 | 1 | 1169 | 1173 | 1162 | 3 |
| rect-fine | 625 | 1183 | 1184 | 1182 | 1 | 1179 | 1182 | 1176 | 2 |

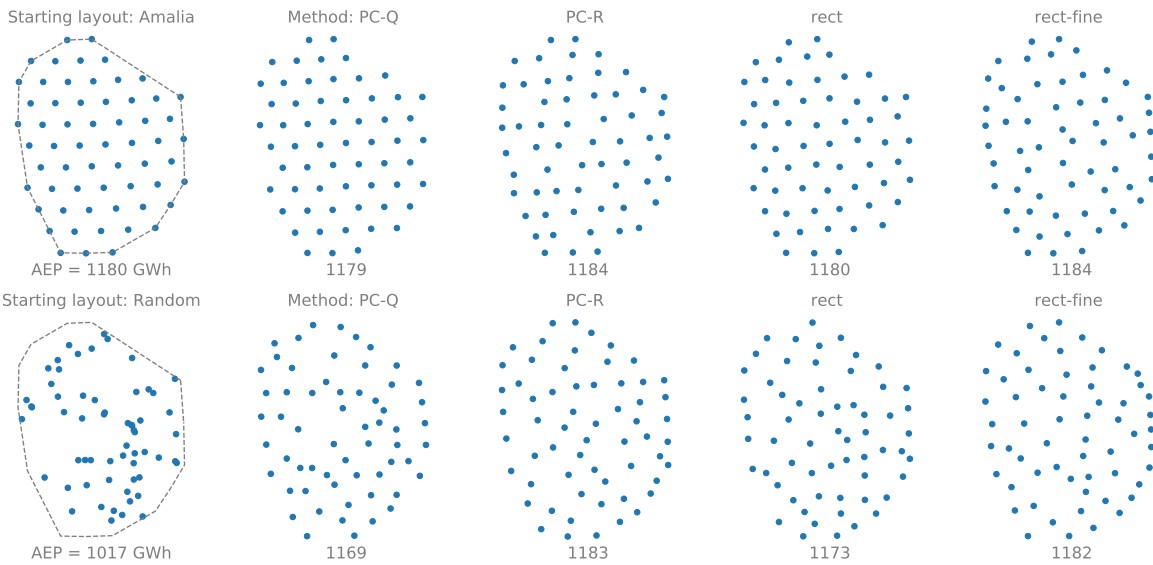

**Figure 10.** Optimal wind farm layouts achieved for each method to compute the AEP. The layouts in the first column show the turbines starting position for the optimization along with the boundary constraint (dashed-line). The optimal layouts correspond to those with the maximum AEP from Table 2. The first row shows the optimums obtained starting from the Amalia layout; we see that the distribution of the turbines is similar to the one of the starting layout. The second row shows the optimums starting from the Random layout; we see that the optimal layouts obtained are less grid-like and produce more novel layouts. For each method, the optimal layouts starting from the Amalia have a higher AEP than those from the Random.

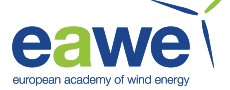
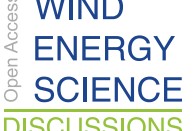

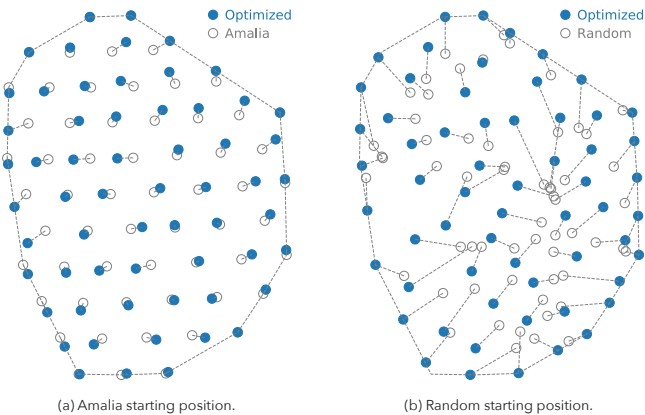

**Figure 11.** The initial and optimized position of the turbines. The turbines starting from the Random layout move more and explore more of the design space. These are the optimum layouts obtained with the PC-R method from Fig. 10.

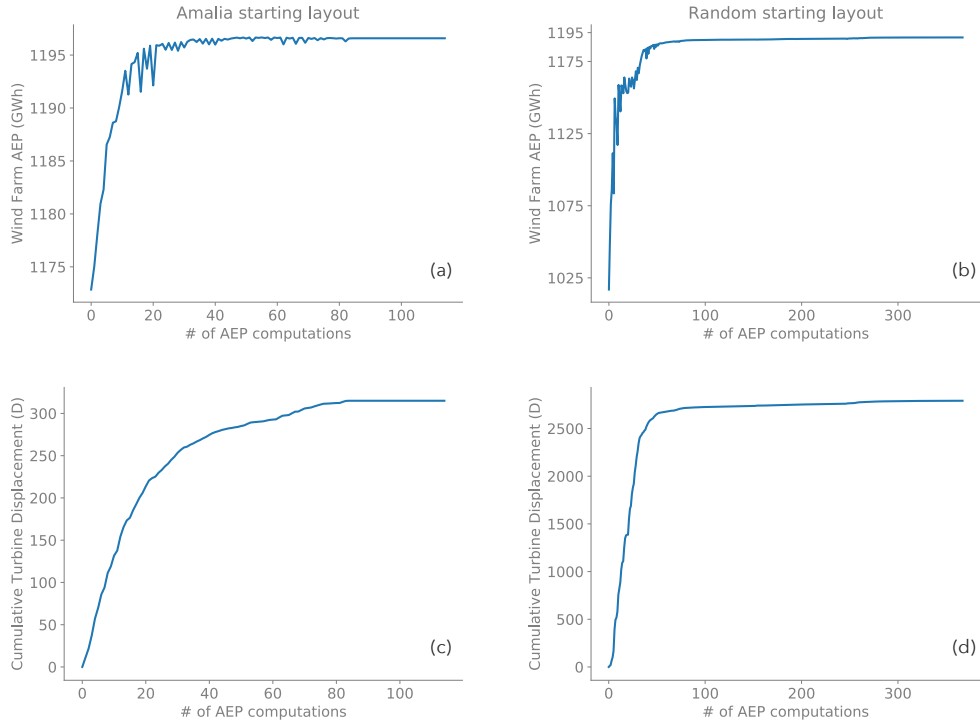

**Figure 12.** Convergence history of the optimization. The top row shows the objective convergence and bottom row shows the design variable convergence. The left column is for the optimization with the Amalia starting position, and the right column is for the optimization with the Random starting position (note the different scales). The optimization of the Amalia starting layout converges in fewer AEP computations. The final AEP for both starting layouts is similar. The turbines move more and thus explore more of the design space for the Random starting layout.





**Table 3.** The improvement in AEP of the optimized layouts from Fig. 10 over their starting layouts. The improvements are smaller when measured by 200,000 Monte Carlo samples rather than directly by the method used to compute the AEP in the optimization.

| Method | Starting layout: Amalia | | Starting layout: Random | |
|---|---|---|---|---|
| | AEP % improvement as measured | | AEP % improvement as measured | |
| | by Monte Carlo | by method | by Monte Carlo | by method |
| PC-Q | -0.09 | 1.23 | 14.91 | 21.34 |
| PC-R | 0.35 | 2.02 | 16.34 | 17.18 |
| rect | 0.02 | 1.35 | 15.33 | 17.75 |
| rect-fine | 0.30 | 2.36 | 16.23 | 17.22 |
| average | 0.14 | 1.74 | 15.70 | 18.37 |

with a first-order finite difference, 120 (the number of design variables) times as many wind farm wake model calls would be needed per optimization. The optimization starting from the Amalia layout (Fig. 12a) converges faster than that starting from the Random layout (Fig. 12b). The optimization with the Amalia starting layout converges in 114 optimization iterations as opposed to 367 for the Random starting layout. The design variables (turbines' positions) change an order of magnitude less

for the Amalia starting layout (Fig. 12c) than for the Random starting layout (Fig. 12d). For the Amalia case, the cumulative displacement of all turbines measured in turbine diameters is 315 and for the Random case 2790. In general, most optimizations we performed behaved similarly to those shown in Fig. 12. The optimizations starting from the Amalia layout run for fewer iterations (usually on the order of 100) with less design variable change than those starting from the Random layout. Most of the optimizations we ran either successfully converged or were on their way to satisfying the convergence criteria before they

reached the time limit we set to save computational cost.

To properly compare the results obtained by the different methods used to compute the AEP in the optimization, we should compute the AEP of the optimal layout with a method that was not used for the optimization. We use 200,000 Monte Carlo samples—wind direction, wind speed pairs—to test the AEP of the optimal layouts. For the optimal layouts in Fig. 10, with the Amalia starting layout, the average improvement in AEP over the starting layout for all the methods is 0.14 % when measured

with the Monte Carlo samples and 1.74 % when measured by the method used in the optimization. For the optimizations starting from the Random layout, the average AEP improvements are 15.70 % for Monte Carlo and 18.37 % for the method used in the optimization (see Table 3). The reporting of the AEP computed by Monte Carlo explains why some of the results reported in Table 2 and Fig. 10 have a smaller AEP than its starting layout.

The improvements starting from the Amalia layout are similar to those found by Gebraad et al. (2017) for turbine position

optimization. The large improvements in AEP for the layouts starting from the Random layout show that the optimizer can find good layouts from a bad starting position. Thus, to search for the best layout, we would perform many optimizations with random starts.





From all the optimizations runs, we conclude that computing the AEP with polynomial chaos based on regression produces the best layouts—around 1 % higher AEP when starting from a random start—for the same number of samples. This is because PC-R computes the AEP more accurately than the other methods for the same number of samples (Sect. 6.2). Furthermore, the optimization with PC-R finds optimums comparable to those found with the rectangle rule that used roughly three times

as many simulations[5]. We found that starting an optimization from a good layout will, in general, find better optimums than starting from a random layout, but starting from the random layout can lead to novel layouts and possibly better layouts as the turbines explore more of the design space. Finally, to properly compare between methods an ensemble of optimizations should be used and evaluated in the same way.

## 7    Conclusions

The application we considered was the layout optimization of a wind farm. The goal of the optimization was to maximize the Annual Energy Production (AEP) of the wind farm. The layout optimization is an optimization under uncertainty (OUU) problem (Fig. 1c) with the wind speed and wind direction as uncertain variables and the positions of the wind turbines as design variables. The AEP was computed with the uncertainty quantification method of polynomial chaos (PC). We tailored the use of PC and extended PC to reduce the number of high-fidelity simulations needed to accurately compute statistics, such as the

AEP, and the gradient of those statistics.

We have shown that polynomial chaos based on regression (PC-R) can compute the AEP accurately using up to an order of magnitude fewer simulations than the rectangle rule, the method currently used in practice in the wind industry. For the case when the AEP is a function of two uncertain variables, wind direction and wind speed, polynomial chaos based on regression computes the AEP accurately (error less than 1 %) usually with only two hundred simulations or less depending on the wind

farm layout—a significant improvement over the current industry practice of using more than a thousand model evaluations to compute the AEP.

The layout of the wind farm influences the convergence of the AEP because the layout has a significant effect on the power output of the farm as the wind conditions vary. We considered four representative layouts: Grid, Amalia, Optimized and Random. The power response of the grid-like layouts (Grid and Amalia) has large oscillations caused by the large drops in

power that occur when rows of wind turbines are aligned with particular wind directions. Because of the larger variability in the power response, the grid-like layouts require more simulations than the non-grid-like layouts (Optimized and Random) to converge the AEP, independent of the method used to compute the AEP.

We extended polynomial chaos to obtain gradients of the statistics, such as the AEP, from the gradients of the power at the simulation samples. Making use of the gradients of the AEP, we performed multiple wind farm layout optimizations to compare

the optimization results obtained with the different methods to compute the AEP: polynomial chaos based on regression, polynomial chaos based on quadrature, and the rectangle rule. We showed that to properly compare methods an ensemble of

---

[5]The rectangle rule used almost three times as many simulations to compute the AEP for each optimization iteration (see the number of samples in Table 2), which results in the optimization using roughly three times as many simulations.



optimizations should be used. Also, for proper comparison, the AEP of the optimized layouts should be evaluated in the same way and with a method different from the one used in the optimization. To be confident about the differences between the optimal layouts, we evaluated the AEP of each optimized layout using the same 200,000 Monte Carlo samples. We found that the benefits of being able to efficiently compute the AEP with PC-R translate to being able to find better optimums than those

5   obtained when computing the AEP with the rectangle rule when using the same number of simulations. With PC-R we find optimums that are 1 % better than those found with the rectangle rule. This is a significant improvement as a 1 % increase in the AEP for a modern large wind farm can increase its annual revenue by millions of dollars.

*Competing interests.*   The authors declare that they have no conflict of interest.

*Acknowledgements.*   Funding for this research was provided by the National Science Foundation under a collaborative research grant No.

10   1539384 and 1539388.





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
