# Peer review of "Polynomial chaos to efficiently compute the annual energy production in wind farm layout optimization"

_Wind Energy Science, 2017_

## Referee Comment (RC1) · Anonymous Referee #1 · 27 Feb 2018

Nice paper - the method could improve the efficiency of wind farm optimization.

Two simple questions, that might be addressed in the paper:

- What if the wind speed does not nicely follow a (single) Weibull distribution? Is there a specific added complexity that might render the method less efficient than the rectangle method?

- Do you think complex constraints, e.g. water depth, wake-induced loads, shipping routes (for support vessels), could be more efficiently handled with the new method?

---

## Author Comment (AC1) · 10 Mar 2018

RC1 - Question 1: "What if the wind speed does not nicely follow a (single) Weibull distribution? Is there a specific added complexity that might render the method less efficient than the rectangle method?"

No, the method would not be less efficient than the rectangle method. Where, by efficient, we mean the number of simulations required to accurately compute the AEP or a statistic of interest. However, there is an added complexity by not having a single Weibull distribution. The added complexity is that we cannot consider the uncertain variables of the wind speed and wind direction to be independent, which is often the

case as they are usually correlated. This added complexity introduces some upfront costs of dealing with the correlated variables, but it would not significantly affect the number of simulations required to accurately compute the AEP. There are different approaches to use polynomial chaos when the input variables are correlated (we cannot fit a single Weibull distribution to the wind speed): 1. Perform a variable transformation to uncorrelate the variables. 2. Construct polynomials that are orthogonal to the multivariate distribution instead of orthogonal polynomials for each dimension. 3. Find subsets of the wind direction where you can fit a single Weibull distribution, and then combine the subsets with what is known as multi-element polynomial chaos.

RC1 - Question 2: "Do you think complex constraints, e.g. water depth, wake-induced loads, shipping routes (for support vessels), could be more efficiently handled with the new method?"

Yes, If the constraints are formulated probabilistically. If the constraints are deterministic, then there would be no difference.

---

## Referee Comment (RC2) · Anonymous Referee #2 · 12 Mar 2018

The article concerns application of polynomial chaos to estimating average energy production (AEP) of a wind-farm, subject to uncertain wind-speed and direction (2 parameter UQ). This PC estimate is used in optimisation under uncertainty, to maximize AEP subject to farm-layout.

The setup and application of both the wind-farm model, and the UQ is competent and clear. Discussion is concise and unambiguously presented, conclusions are well founded based on results, and the presentation is professional.

My main concern is that this work may not be innovative enough. Certainly from a UQ perspective, the methods applied are perhaps the *default* UQ methods, applied in the

standard way, and they in fact seem not very well suited to this particular problem. The problem has many features that make it a challenging and unique UQ problem: dependent input uncertainties, non-smooth distributions and responses, periodic parameters, significant noise, need for smoothness in the turbine-position space (to aid optimization). None of which are adequately addressed in the choice of numerical methods or the discussion.

The authors mention that the rectangle rule is standard in wind-energy, and PC is largely unknown. I would say this is article not a great advert for PC because of the mediocre results - but on the other hand if this is one of the first applications of modern UQ to this problem, I could see the value. I recommend major revisions:

Major comments:

- PC is interpolating/regressing figure 5, the power as a function of wind-speed (y) and direction (x), which shows a highly irregular pattern in the x. I suggest polynomials may be quite a poor choice for approximating this function. This could be verified by the authors if they plotted the implied response surfaces of PC-R and PC-Q and compared with this reference - oscillations may be present, as well as high sensitivity to the sample locations (hence perhaps their 10 runs with varying samples). In contrast the rectangle rule is just "pixellating" Fig 5. Given the periodicity of and shape of x, I would use combination of a Fourier series in x, and a polynomial in y. An equivalent integral approximation can be built, and since the underlying representation better matches the response, the AEP should be better.

- The fact that PC is perhaps not a good choice here, is additionally suggested by that fact that it performs only very slightly better than the rectangle rule (I don't agree with the authors interpretation of significant improvements in Figures 6-8). Rectangle should be 2nd-order while PC should be spectral. I suggest the lower variance of PC-R in Fig 9 is most likely the effect of PC-R filtering noise with regression. PC is likely not significantly improving the representation of Fig 5, compared to the under-sampling of

the rectangle rule.

- The authors should not underestimate the effect non-independence of wind-speed and direction may have on the AEP. In my experience (in unrelated problems) dependence relationships in inputs are significantly more important than non-Gaussianity (skewness, kurtosis, etc.) of 1d-marginals. This makes the careful choice of Weibull potentially irrelevant for the purposes of comparing layouts. Please plot the 2d distribution in Fig 3, so we can see how strong the dependence is. Mention how PC could be generalized to allow for this (there is some literature on the subject). Computing the effect of this on the AEP would also be a very nice addition.

- Justify why wind-speed is fit with a distribution, but direction not.

- Justify why computing time of this problem is relevant. This is a one-off optimization for a farm that might last 20 years.

- Given your results it seems that the layout problem could have a very large subspace of close-to-optimal designs - all essentially equivalent. Do you agree? Please comment.

Minor comments:

- I have a personal interest in wake-deflection, which is mentioned in connection with FLORIS. Could the authors comment on how the layout problem would change if optimal wake deflection were allowed?
* * *

---

## Author Comment (AC2) · 15 Apr 2018

Thank you again for your comments and questions. We really appreciate them. We just wanted to follow up on our previous response to inform you that we have now addressed your comments in the paper. Specifically, see the fifth paragraph of the Discussions and conclusions section.

---

## Author Comment (AC3) · 15 Apr 2018

Thank you for taking the time to thoroughly review the manuscript and for your insightful comments. We have updated the manuscript to reflect your comments as detailed below under each specific comment. In general, we included many of the comments in a new Discussions and conclusions (DC) section that greatly expanded our previous Conclusions section.

RC2 - Comment 1: "PC is interpolating/regressing figure 5, the power as a function of wind-speed (y) and direction (x), which shows a highly irregular pattern in the x. I suggest polynomials may be quite a poor choice for approximating this function. This

could be verified by the authors if they plotted the implied response surfaces of PC-R and PC-Q and compared with this reference - oscillations may be present, as well as high sensitivity to the sample locations (hence perhaps their 10 runs with varying samples). In contrast the rectangle rule is just "pixellating" Fig 5. Given the periodicity of and shape of x, I would use combination of a Fourier series in x, and a polynomial in y. An equivalent integral approximation can be built, and since the underlying representation better matches the response, the AEP should be better."

We agree that one could potentially build a better approximation in the x-direction, especially using more samples, but we are not sure that for the same number of samples it could provide a better estimate of the AEP. We've included a couple of sentences with your suggestions and to encourage future research to look into other approximations at the end of the third paragraph of the new Discussions and conclusions (DC) section. One of our goals of this work is to encourage people to look into better ways to approximate the AEP than the rectangle rule, and we will be happy to see others try to improve the results shown here.

Also, we have included a paragraph (paragraph 4 of DC) discussing why polynomial chaos is a good choice for this problem. We have also emphasized the fact that we show how to compute the gradients of the statistics with polynomial chaos. The gradients are novel for the case of the polynomial chaos based on regression. These changes have been made in a couple of places including modifying a sentence in the abstract and adding a sentence in the introduction, among other places (see the pdf marked up with the differences).

During our work, we have visualized the polynomial approximations and the rectangle rule approximations to the power response (figure 5), they indeed do not exactly match the power responses shown in figure 5. However, the approximations do get better as the number of samples used to construct the approximation increases, as can be seen in figure 8. The power response is a complicated function, and the approximations we build with only a couple hundred samples are not going to be able to capture all the

high-frequency oscillations, but they can give good estimates of the AEP as we have shown. For reference, Figure 5 was constructed using 32,400 samples (a grid made by 360 wind directions (every degree) and 90 wind speeds). Also, as mentioned in the paper, even using 200,000 MC samples to estimate the AEP only ensures with 99 % confidence that the true AEP is within $\pm$ 1 % of estimated AEP.

RC2 - Comment 2: "The fact that PC is perhaps not a good choice here, is additionally suggested by that fact that it performs only very slightly better than the rectangle rule (I don't agree with the authors interpretation of significant improvements in Figures 6-8). Rectangle should be 2nd-order while PC should be spectral. I suggest the lower variance of PC-R in Fig 9 is most likely the effect of PC-R filtering noise with regression. PC is likely not significantly improving the representation of Fig 5, compared to the under-sampling of the rectangle rule."

We believe that reductions of up to an order of magnitude in the number of simulations are significant. As mentioned above, we have included a paragraph (paragraph 4 of DC) discussing why polynomial chaos is a good choice for this problem, in addition to that of computing the AEP more efficiently than the rectangle rule. You are correct in your observation of Fig 9. We have added this to the caption of Fig 9.

RC2 - Comment 3: "The authors should not underestimate the effect non-independence of wind-speed and direction may have on the AEP. In my experience (in unrelated problems) dependence relationships in inputs are significantly more important than non-Gaussianity (skewness, kurtosis, etc.) of 1d-marginals. This makes the careful choice of Weibull potentially irrelevant for the purposes of comparing layouts. Please plot the 2d distribution in Fig 3, so we can see how strong the dependence is. Mention how PC could be generalized to allow for this (there is some literature on the subject). Computing the effect of this on the AEP would also be a very nice addition."

We have made it clearer that we constructed the wind direction and wind speed to be independent. We use the NoordZeeWind meteorological mast data, which is dependent, as inspiration to create the probability distributions (see the modifications in 5.1 Probability distributions of the uncertain wind conditions). In the paper, we provide a reference that shows a plot of the experimental correlated data we used as a base to construct the independent distributions.

We have added a paragraph (paragraph 5 of DC) discussing different methods to generalize PC for dependent variables, and we have stated that an interesting extension to the paper would be to study the effects of different wind distributions, including correlated distributions, on the convergence of the AEP.

RC2 - Comment 4: "Justify why wind-speed is fit with a distribution, but direction not."

Both, the wind speed and wind direction distributions are specified by histograms containing 50 bins of equal width. We have clarified this in the updated section 5.1 Probability distributions of the uncertain wind conditions. We created the histogram data for the wind direction by linearly interpolating the experimental data. For the wind speed, we did a two-step approach. First, we fitted a Weibull to the experimental data, then we truncated the Weibull and used its likelihood values to create the histogram. We fitted the Weibull to smooth the data and because this is commonly done for the wind speed (we have included a couple of citations referring to this in section 5.1 as well). Granted, we could have constructed the speed histogram without fitting the Weibull, but we followed the wind industry standard of fitting the wind speed with a Weibull.

RC2 - Comment 5: "Justify why computing time of this problem is relevant. This is a one-off optimization for a farm that might last 20 years."

The final optimization would be a one-off optimization. However, to get to the final optimization, many optimizations are required during the design phase. For example, designers may need to explore scenarios with different turbine types, different sites, larger farms with a different number of turbines and possibly even systems of wind farms. Also, the presence of local optima would require many optimizations with different restarts to find the best layout. And, furthermore, there is a desire to increase the

fidelity of the models used to simulate the wind farm, which will increase the time and computational cost of the optimizations. Thus reducing the computational time is relevant as it will enable the use of higher fidelity models and also facilitate the exploration of many different and larger wind farm designs. In the paper, we have included this justification of why the computing time is relevant in the first paragraph of the Discussions and conclusions section.

RC2 - Comment 6: "Given your results it seems that the layout problem could have a very large subspace of close-to-optimal designs - all essentially equivalent. Do you agree? Please comment."

We agree. We have commented on this in the new Discussions and conclusions (DC) section. Specifically, at the end of paragraph 6 of DC section.

RC2 - Comment 7: "I have a personal interest in wake-deflection, which is mentioned in connection with FLORIS. Could the authors comment on how the layout problem would change if optimal wake deflection were allowed?"

We have commented on this in the last paragraph of the Discussions and conclusions section.
* * *

---

## Author Response (AR2)

Dear Associate Editor,

Thank you for your comments. It is unfortunate to hear that one of the reviewers is unsatisfied with our changes.

We have clearly presented our results and showed that using Polynomial Chaos to compute the AEP is better than the rectangle rule, albeit the results are not as good as the reviewer expected. It is true that the convergence results for PC should be exponential and for the rectangle rule second-order, but these are theoretical results observed in the limit (usually demonstrated in toy problems with analytic solutions and using an impractical number of model evaluations). The computation of the AEP is a challenging problem especially because of the highly oscillatory and non-smooth responses; thus, for the practical number of samples considered, we do not observe the theoretical convergence of the methods. Instead, we focus on the practical aspect of performing wind farm layout optimization, and for this, we have shown that PC is better as it can compute the AEP to a given accuracy with fewer samples than the rectangle rule and also find better optimal layouts. We believe this is a valuable and useful insight to practitioners.

The reviewer brings up a good point about using Fourier series. Due to the oscillatory response of the power with respect to wind direction, it could be beneficial to compute the AEP by a Fourier approximation which is expected to converge exponentially (similarly to PC's theoretical convergence rate). In our paper, we hope to motivate moving beyond the use of the rectangle rule to compute the AEP. We have shown why the Polynomial Chaos could be a good method to replace the rectangle rule not just to compute the AEP but for the optimization problem in general. We have not said it is the best method for computing the AEP, especially as custom integration methods could be built to approximate the AEP. Methods making use of Fourier series would be an option. We believe other researchers should investigate these different integration methods. In the paper, we can say more about other methods to compute the AEP if that would be helpful. And be clearer saying that PC is an approach and not necessarily the best approach to compute the AEP.

Yes, the improved performance of PC-R comes because it is a smoother approximation to the power response. It is likely that the benefits from using Fourier series would also be because it smooths out the response. We had mentioned the smoothing behavior of PC in the paper before the reviewer comment, following the reviewer comment we made this point clearer by reiterating it in Figure 9's caption as well to make sure it is not overlooked.

The benefits of PC-R come from smoothing the response, not the input distribution. In fact, we observed the benefits of PC-R over the rectangle rule for smooth input distributions as well. We had initially performed the study on a smoothed wind distribution and also on a uniform distribution for the wind direction, and for both PC-R performed better. We went with the non-smooth wind direction distribution as it was the most realistic. We took great care in not treating the methods differently and in not making one method look better or worse.

In conclusion, we have clearly illustrated the application of modern UQ methods for the computation of the AEP and for the larger wind farm optimization problem. We have shown that Polynomial Chaos is better than the methods commonly used. As this is one of the first applications of UQ to the wind farm optimization problem, we do expect there would be many ways including those in the reviewer's comments to improve the work presented here. And, we believe looking at different methods to improve our work should be pursued in new papers, instead of significantly changing the work we have presented.

Thank you for your feedback, and we look forward to hearing your thoughts.

Regards,
Santiago

---

## Author Response (AR3)

Thank you for taking the time to review the manuscript and for your comments. We have updated the manuscript to reflect all of your comments as detailed below.

*In the manuscript the authors introduce methods common in the uncertainty quantification (UQ) community to the problem of determining the optimal layout of wind farms. The manuscript is nicely written and the results are clearly presented and explained. As far as I know this is one of the first applications of UQ methods to this problem and therefore the manuscript has a clear added value for the community.*

*After reading the manuscript I had the impression that the authors are "overselling" how well the polynomial chaos approach works to optimize the Annual Energy Production (AEP) of windfarms when compared to the more traditional methods. The authors use 200,000 Monte Carlo simulations to evaluate the performance for each layout and they report that this results in a 1% uncertainty. However, the AEP obtained using polynomial chaos theory is only about 1% higher than using the traditional methods (I am not arguing that a 1% improvement is not important as it clearly is). But the question arises whether the obtained improvement is actually statistically significant and consistent (is it just obtained for this test case or also for other testcases?). Some additional analysis could better solidify this point.*

To ensure our results are consistent, we have re-run all the convergence and optimization studies with the wind speed upper limit set at 25 m/s (the turbine's cut-out speed). In addition, to avoid any potential uncertainty introduced by the MC samples, we used the same 200,000 MC samples (wind direction, wind speed pairs) to evaluate the optimal layouts obtained by each method. Also, for the optimization, we considered three starting layouts (as opposed to two). And we considered optimizations using polynomial chaos and the rectangle rule with both a fine and coarse set of samples. And as before, for each combination of the method to compute the AEP and starting layout, we perform ten optimizations.

The conclusion from this extended new set of results stays the same: polynomial chaos is better than the rectangle rule.

To address the "overselling" how well the polynomial chaos works, we now always report improvements on average over all cases (layouts,

number of samples), before we had considered improvements still on average (over 10 realizations) but for particular cases, like optimizations with a particular starting layout.

We have significantly updated the section with the optimization results, and all the figures and tables in that section (Sect 6.3.2).

*Related to emphasizing the 1% the authors claim while using the polynomial chaos theory they mention, but do not discuss too much, that the result obtained using the actual Amalia layout as starting position is about 0.62% better than using a random starting layout. Should the introduced optimization method not reach at least the same power production for the optimal case as obtained using the Amalia layout as starting position? Now one gets the impression that the method that was used to set the Amalia wind farm layout is even better than the approach discussed here. As a different cost function is used in the present study than the one used to set the Amalia wind farm layout it is of course obvious that a different optimal layout is obtained in this study.*

The problem has many local optima. Of the more than 100 optimizations we ran, every single optimization converged to a different local optimum. Figure 10 shows a subset of the different local optima found. The Random layout was bad to start (some of the turbines are clustered very tightly and almost on top of each other), so the optimizer can end up finding not-so-good local optima. Starting from a good layout will usually result in a better local optimum than starting from a bad layout, which helps explain the better performance of the optimal layouts found starting from the Amalia than those starting from the Random layout. We've added some discussion on this in the rewritten optimization results section.

For the new results, we considered a new random layout that was required to satisfy some minimum spacing constraints between the turbines. In this case, the optimums starting from the Amalia layout are on average 0.2% better than those starting from the Random layout. We also consider the Grid layout as a starting layout as discussed in the optimization results section.

*Based on these points above one gets the impression that indications that the polynomial chaos approach may be better are really emphasized, while potential shortcomings are not discussed at equal footing.*

*Some additional questions are:*

*1. Would the introduced method be able to handle correlated uncertain variables? Now the authors decided to take wind speed and wind direction as uncorrelated and this may influence the results.*

Yes, the polynomial chaos method is able to handle correlated uncertain variables. We have added subsection 4.4 in the polynomial chaos section discussing in detail how to handle correlated uncertain variables.

*2. In section 5.1 the authors set an upper limit for the wind speed of 20 m/s. When I look at figure 5 the different wind farm configurations do not necessarily reach full production (320 MW) for each wind direction at this wind speed. How does this influence the result?*

We have re-run everything with the upper limit set at 25 m/s (the cut-off speed of the turbine). As shown in the results the conclusions stay the same. Of course, the AEP values are higher now.

*3. In the present study the authors only consider wind speed and direction to be uncertain. As the authors rightfully argue that there can be much more uncertain variables when considering wind farm design? How will the polynomial chaos approach in comparison to other methods when there are much more uncertain variables one wants to consider?*

The polynomial chaos method will continue to perform better until 5-10 uncertain variables when the Monte Carlo method will start to perform better because it does not suffer from the curse of dimensionality. The curse of dimensionality---the sampling requirements for the rectangle rule and polynomial chaos increase exponentially with increasing dimension. We've added a discussion on this in the Discussion and Conclusions section. Also, we've added some more details in the polynomial chaos section.

*4. The authors mention in the reply to referee 2 that the number of model evaluations is reduced by a factor of 10 when the polynomial chaos method is used. However, it is unclear to me on what this statement is based. Even in figure 6-8 I struggle to see such a large improvement, maybe apart from some specific cases. When we look at one of the actual test cases (table 2) we see that for the same number of iterations the polynomial chaos indeed gives a better result than using the rectangle rule. However, the results obtained using the rectangle rule using less than 3 times the number of simulation is already better (so not 10 times more simulations). Again this gives the impression the authors are overselling the benefits of the polynomial chaos method too much.*

As mentioned above, we now always report improvements on average over all cases. We've updated the abstract, conclusions and results section with the updated improvements numbers.
With regards to the factor of ten, it was referring to computing the AEP to a certain accuracy for one of the layouts. The up to a factor of ten was not meant for the optimizations.

*My overall impression is that the work provides valuable contribution to the field of windfarm optimization. The problem is very challenging and the paper shows that polynomial chaos approach may be a good additional tool to do this; and combinations of different optimization approaches are particularly useful for this problem. However, it would be nice when the authors can indicate (or extend when possible) the approach to include correlated dependent variables, the effect of setting an upper limit for the wind speed of 20 m/s, and how the method would work when there are more independent variables. And it would be particularly important to give a more balanced summary of the strengths and potential drawbacks of the polynomial chaos approach as outlined in my report above.*

Thanks again for your comments. As discussed above, we have incorporated your suggestions and strived to give a balance summary of the results.

[revised manuscript text omitted]

---

## Author Response (AR4)

Thank you for your suggestions. We have incorporated them in the final manuscript has detailed below. The specific changes can be seen in the difference pdf file.

*In the updated version the authors have addressed most of my suggestions / comments from my previous report. Based on the changes, and after reading the manuscript again, I have the following final suggestions with respect to this manuscript. When these are incorporated the*

*End section 6 and discussion conclusion section*
*\* Here, the authors discuss the results obtained using the different starting layouts. For example, it is found that the optimal layout obtained using the Amalia layout as starting configuration is not as good as using a rectangular grid. It should be added that the Amalia layout has been obtained using different optimization constraints than used in the present study (or simply by using a slightly different wake mode). Given the local maxima in the optimizations procedure this causes the optimization algorithm to get stuck in this local optimum that is created.*

We have noted that the Amalia layout has been obtained by solving a different optimization problem as shown in the diff pdf file.

\* In the abstract the authors should also add a sentence on the sensitivity of the starting configuration, which clearly is also an important consideration in this optimization.

We have incorporated this suggestion in the abstract.

\* "With PC based on regression, we have reduced by a factor of five the number of simulations required to accurately compute the AEP, thus enabling the use of more expensive, higher-fidelity models or larger wind farm optimizations."
==> Please specify reduced by a factor of five compared to "???"
==> Later on in the abstract a reduction factor of 3 is mentioned. So make sure what the factor 3 and 5 exactly refer to.

We have rewritten the second half of the abstract to incorporate both of the above suggestions, i.e., we have added a sentence on the sensitivity of the starting configuration and mentioned exactly what the factor of 3 and 5 refer to.

\* in the abstract and conclusion specify that the numbers have been obtained for this one wind farm test case.

We have explicitly specified in both the abstract and the conclusion that the number obtained have been for the cases considered as can be seen in the diff pdf file.

Minor:
*Around equation 36 there is a sentence: "36 wind directions the chosen intervals are {[0, 360]; [10, 370]; [20, 380]; . . . ; [340, 700]; [350, 710]"
==> The mentioned wind angle ranges are somewhat strange

We've change this sentence to make it less confusing and easier to understand.

* Section 6 first sentence
AEP has already been introduced before.

Thanks, we have deleted this sentence.

*Section 6.2 In what follows we will compare the PC-R (the best performing method) with the rectangle rule (the method currently used in practice)
==> please formulate more generally; the rectangle rule is not the only method used

We have changed it to the more general statement of "to the method most commonly used in practice." In section 3, when describing the methods, we mention other methods besides the rectangle rule.

[revised manuscript text omitted]